# Three F-actin assembly centers regulate organelle inheritance, cell-cell communication and motility in *Toxoplasma gondii*

**Nicolò Tosetti, Nicolas Dos Santos Pacheco, Dominique Soldati-Favre\*, Damien Jacot\***

Department of Microbiology and Molecular Medicine, CMU, University of Geneva, Geneva, Switzerland

**Abstract** *Toxoplasma gondii* possesses a limited set of actin-regulatory proteins and relies on only three formins (FRMs) to nucleate and polymerize actin. We combined filamentous actin (F-actin) chromobodies with gene disruption to assign specific populations of actin filaments to individual formins. FRM2 localizes to the apical juxtanuclear region and participates in apicoplast inheritance. Restricted to the residual body, FRM3 maintains the intravacuolar cell-cell communication. Conoidal FRM1 initiates a flux of F-actin crucial for motility, invasion and egress. This flux depends on myosins A and H and is controlled by phosphorylation via PKG (protein kinase G) and CDPK1 (calcium-dependent protein kinase 1) and by methylation via AKMT (apical lysine methyltransferase). This flux is independent of microneme secretion and persists in the absence of the glideosome-associated connector (GAC). This study offers a coherent model of the key players controlling actin polymerization, stressing the importance of well-timed post-translational modifications to power parasite motility.
DOI: https://doi.org/10.7554/eLife.42669.001

**\*For correspondence:**
Dominique.Soldati-Favre@unige.ch (DS-F);
damien.jacot@unige.ch (DJ)

## Introduction

The large phylum of Apicomplexa is composed of thousands of protozoan pathogens of medical and veterinary significance including *Toxoplasma gondii* and the *Plasmodium* species responsible for toxoplasmosis and malaria, respectively (*Adl et al., 2007*; *Seeber and Steinfelder, 2016*). To survive and disseminate, these obligate intracellular parasites have developed complex strategies to invade host cells, replicate inside a parasitophorous vacuole (PV), avoid immune attacks and interfere with host cellular defence mechanisms. In *T. gondii*, generation and dynamics of F-actin are known to be critical for apicoplast inheritance (*Andenmatten et al., 2013*; *Jacot et al., 2013*), constriction of the basal pole, intravacuolar cell-cell communication (*Frénal et al., 2017b*; *Periz et al., 2017*) and gliding motility (*Dobrowolski and Sibley, 1996*; *Drewry and Sibley, 2015*; *Wetzel et al., 2003*) (*Figure 1A*).

Present in most apicomplexans, the apicoplast is a plastid-like, secondary endosymbiotic organelle surrounded by four membranes that hosts essential metabolic pathways (*McFadden et al., 1996*; *McFadden and Yeh, 2017*). During parasite division, the apicoplast segregates between the two forming daughter cells through the action of myosin F (MyoF), a motor conserved across the phylum of Apicomplexa (*Jacot et al., 2013*). Concordantly, actin is necessary for this process in both *T. gondii* and *Plasmodium falciparum* (*Andenmatten et al., 2013*; *Das et al., 2017*). Additionally, MyoF is reported to participate in the trafficking of dense granules (*Heaslip et al., 2016*). Dense granules constitutively secrete dense-granules proteins (GRAs) both into and beyond the PV

**Figure 1.** Schematic representations of *T. gondii* division, motility and invasion. (**A**) Intracellular growth development of *T. gondii* consists of the synchronous geometric expansion of two daughter cells within a mother cell. Apicoplast inheritance is coupled to cell division. All parasites are connected by their basal pole to the central residual body (RB) that allows rapid diffusion of materials between parasites of the same parasitophorous vacuole (PV). The PV contains a network of elongated nanotubules that form connections with the PV membrane. (**B**) Schematic representation of a gliding parasite. The parasite plasma membrane (PPM) and the inner membrane complex (IMC, a system of flattened membranous sacs called alveoli that directly underlies the PPM) compose the pellicle. Transmembrane adhesins (MICs) are secreted apically by the micronemes and will interact with host cell ligands. Within the pellicle MICs bind to GAC with the latter connecting the complex to F-actin. The rearward translocation of the GAC-adhesin complexes by the successive action of the MyoH and MyoA glideosomes will result in parasite forward motion. (**C**) During invasion, rhoptry organelles secrete the rhoptry neck proteins (RONs) in the host plasma membrane. This parasite-derived receptor will interact with the micronemal apical membrane antigen 1 (AMA1) to form the moving junction (MJ). The rearward translocation of this junction by MyoH and MyoA will result in host cell invasion. Invagination of the host plasma membrane leads to the formation of the PV. APR: apical polar ring.

DOI: https://doi.org/10.7554/eLife.42669.002

(*Mercier and Cesbron-Delauw, 2015*). Some GRAs play a role in the structural modifications of the PV including the formation of an intravacuolar membranous nanotubular network (*Mercier et al., 2002*) while other are implicated in subversion of host cell defense mechanisms (*Figure 1A*) (*Bougdour et al., 2013*; *Gold et al., 2015*).

F-actin is also implicated in a unique mode of intravacuolar cell-cell communication (*Frénal et al., 2017b*; *Periz et al., 2017*), which is mediated by myosin I (MyoI) and responsible for the synchronicity of parasite division within a given vacuole (*Frénal et al., 2017b*). A posterior membranous structure called the residual body (RB), where myosin I (MyoI) is located, connects all intravacuolar tachyzoites, allowing the diffusion of proteins and the transport of vesicles between parasites (*Figure 1A*). Finally, the basal pole constriction of the parasites is governed by myosin J (MyoJ), which also participates in the establishment of cell-cell communication (*Figure 1A*) (*Frénal et al., 2017b*).

Gliding motility is a prerequisite for host cell invasion, parasite egress and migration across biological barriers and is powered by the glideosome, a molecular machine conserved across the phylum (*Frénal et al., 2017a*). The glideosome is composed of an actomyosin system that promotes the rearward translocation of transmembrane adhesins attached to the extracellular matrix. During motility, the adhesins are discharged apically at the parasite plasma membrane by regulated secretory organelles called micronemes (*Figure 1B*). In *T. gondii*, these micronemal adhesins (MICs) typically assemble in complexes composed of one transmembrane protein associated with one or more proteins exhibiting host cell binding properties (*Carruthers and Tomley, 2008*). In *Plasmodium*, the adhesins thrombospondin-related adhesive protein (TRAP) and the circumsporozoite-and TRAP-related protein (CTRP) participate in sporozoites and ookinetes motility, respectively (*Dessens et al., 1999*; *Kappe et al., 1999*; *Steinbuechel and Matuschewski, 2009*; *Sultan et al., 1997*). During invasion, parasite-derived receptors, contained in the rhoptry organelles (rhoptry neck proteins, RONs) are apically discharged onto the host plasma membrane. Upon secretion, the RONs assemble with the microneme protein AMA1 (apical membrane antigen 1) to form the moving junction (MJ) (*Besteiro et al., 2011*; *Lamarque et al., 2014*). The basal translocation of this MJ by the actomyosin system propels the parasite inside the host cell, a process conserved in both *T. gondii* and *Plasmodium* (*Figure 1B–C*) (*Riglar et al., 2011*; *Riglar et al., 2016*; *Vulliez-Le Normand et al., 2012*). At the end of the invasion process, the parasite exhibits a twisting motion (*Pavlou et al., 2018*) sealing the PV and creating a safe and secluded niche for replication (*Figure 1C*). In *T. gondii*, the adhesins are connected to the parasite F-actin via the glideosome-associated connector (GAC) (*Jacot et al., 2016*) and their rearward translocation is powered by the successive actions of MyoH and MyoA (*Graindorge et al., 2016*; *Long et al., 2017a*). Parasite motility is initiated at the conoid, an apical protruding organelle composed of tubulin fibers, by the action of MyoH, which presumably translocates F-actin and GAC in the confined space between the plasma membrane and the inner membrane complex (IMC) that compose the pellicle in Apicomplexa (*Graindorge et al., 2016*). MyoA, anchored in the pellicle, takes the relay at the level of the apical polar ring (APR) to translocate the adhesin complexes to the basal pole of the parasite (*Figure 1B–C*) (*Andenmatten et al., 2013*; *Frénal et al., 2014*).

The signaling events leading to parasite egress from infected cells involve the coordinated stimulation of micronemes exocytosis and activation of the actomyosin system. Egress is initiated by the activation of the cGMP-dependent protein kinase (PKG) (*Brown et al., 2017*; *Brown et al., 2017*; *Wiersma et al., 2004*), a central regulator that is involved in phospholipase C (PLC) and calcium-mediated signaling (*Bullen and Soldati-Favre, 2016*; *Jia et al., 2017*; *Lourido et al., 2012*). PLC produces inositol triphosphate (IP$_3$), which presumably opens an unknown IP$_3$-sensitive Ca$^{2+}$ channel (*Garcia et al., 2017*; *Lovett et al., 2002*) and diacylglycerol (DAG), which is converted on the inner leaflet of the plasma membrane into phosphatidic acid (PA) via diacylglycerol kinase 1 (DGK1) (*Bullen et al., 2016*). The acylated plekstrin homology domain-containing protein APH, located on the microneme surface, binds to PA and mediates microneme exocytosis (*Bullen et al., 2016*; *Darvill et al., 2018*). On the other hand, release of Ca$^{2+}$ results in the activation of the Ca$^{2+}$ dependent protein kinase 1 (CDPK1), an essential effector of microneme secretion (*Lourido et al., 2010*). In addition, but only in some conditions, a second kinase, CDPK3, contributes to the process (*Garrison et al., 2012*; *Lourido et al., 2012*; *McCoy et al., 2012*; *Treeck et al., 2014*). Although significant advances have been made to decipher these signaling pathways, it remains unclear which phosphorylation events are required. Furthermore, given the essential contribution of the micronemal adhesins in motility, it has been so far impossible to disentangle the role of signaling factors in the activation of the actomyosin system from microneme exocytosis. In contrast to CDPK1-dependent phosphorylation, lysine methylation mediated by the apical lysine methyltransferase (AKMT) selectively affects motility without impairing microneme secretion (*Heaslip et al., 2011*). Remarkably, AKMT plays a critical role in recruiting GAC at the apical tip of the parasite (*Jacot et al., 2016*).

F-actin has notoriously been difficult to detect in apicomplexans and actin was assumed to be maintained predominantly as a large pool of monomers (*Dobrowolski et al., 1997*; *Mehta and Sibley, 2010*; *Olshina et al., 2012*; *Skillman et al., 2011*). Recently, actin chromobodies (Cb) allowed the detection of F-actin networks mainly accumulating in the RB of *T. gondii* (*Periz et al., 2017*). All the apicomplexans lack the ARP2/3 complex (*Gordon and Sibley, 2005*), and *T. gondii* relies on three formins (FRMs) to nucleate and polymerize actin. FRM1 and FRM2 are well conserved across

the phylum, whereas FRM3 is restricted to the subgroup of coccidians and shown before to be dispensable (*Daher et al., 2012*). Although FRM1 and FRM2 were both localized at the pellicle using antibodies raised against bacterially produced FH2 domains (*Daher et al., 2010*), endogenous epitope-tagging of both FRMs revealed different localizations. FRM1 is restricted to the apical tip of the parasite (*Jacot et al., 2016*), a localization more concordant with the apical localization of *P. falciparum* FRM1 (*Baum et al., 2008*; *Douglas et al., 2018*) and its predicted role in generating F-actin apically to initiate motility (*Graindorge et al., 2016*; *Jacot et al., 2016*; *Long et al., 2017a*).

Here, endogenous epitope–tagging was used to localize FRM2 to an apical juxtanuclear region, while FRM3 is present at the basal pole and RB. To assign F-actin-dependent processes to each individual formins, we took advantage of combining F-actin chromobodies imaging (*Periz et al., 2017*; *Tardieux, 2017*) with a series of gene disruptions. We establish that in intracellular parasites, two distinct populations of F-actin accumulate mainly to the apical juxtanuclear region and to the RB. FRM2 is responsible for the generation of F-actin in this apical juxtanuclear region and participates in apicoplast inheritance, while FRM3 produces F-actin in the RB to ensure synchronicity of division by maintaining cell-cell communication. In contrast, FRM1 produces actin filaments that are only visible in extracellular parasites to enable gliding, invasion and egress. These filaments translocate along the pellicle by the successive actions of MyoH and MyoA and accumulate at the basal pole even in the absence microneme secretion or GAC. This flux is triggered by PKG and calcium signaling cascades and requires CDPK1 and AKMT, indicating that both phosphorylation and methylation coordinate the regulation of the actomyosin system.

## Results

### FRM2 localizes to the apical juxtanuclear region and participates in apicoplast inheritance

A stable line expressing GFPTy fused to F-actin chromobodies (Cb-GFPTy) was generated in wild type (wt) RH parasites. Cb-GFPTy showed strong staining of the RB as previously reported (*Periz et al., 2017*), and in a juxtanuclear region often overlapping with the apicoplast (*Figure 2A*). C-terminally Ty-tagged FRM2 at the endogenous locus was found restricted to the same apical juxtanuclear region (*Figure 2B*). In non-dividing parasites, FRM2 always co-localized with the Golgi and only transiently overlapped with the apicoplast (*Figure 2C*). Even in the absence of the apicoplast, which can be chemically removed using high doses of anhydrotetracycline (ATc, 4 μg/ml) (*Jacot et al., 2013*), FRM2 remained localized to the juxtanuclear region (*Figure 2—figure supplement 1A*). During *T. gondii* tachyzoites division, the apicoplast associates with the duplicated centrosomes to ensure its encapsulation in the growing daughter cells (*Striepen et al., 2000*; *van Dooren et al., 2009*). Concordantly, FRM2 was found concentrated at the edges of the elongating and dividing apicoplast (*Figure 2D* and *Figure 2—figure supplement 1B*), where the two centrosomes are positioned. Endogenously tagged FRM2 in a MyoF inducible knockdown (MyoF-iKD) strain confirmed co-localization of the two proteins in this juxtanuclear region and depletion of MyoF did not affect FRM2 localization (48 hr +ATc) (*Figure 2E*). To assess FRM2 function, two independent *FRM2* knockout mutants were generated in RH parasites using a two-gRNAs CRISPR/Cas9 approach, resulting in the deletion of a large part of the open reading frame (*Figure 2—figure supplement 2A*). FRM2-KO parasites exhibited a defect in apicoplast inheritance (*Figure 2F*), while the segregation of the Golgi was unaffected (*Figure 2—figure supplement 2B*). The phenotype was partial; with about ~30% of vacuoles displaying a proper distribution of apicoplast in all the parasites,~50% of the vacuoles showing at least one parasite lacking the apicoplast, and ~20% with all the parasites lacking the apicoplast (resulting from the reinvasion of parasites lacking the apicoplast) (*Figure 2G*). Consistently, FRM2-KO parasites survived but exhibited a significant defect in growth competition assay against a GFP expressing wt strain, while the parental RH strain was not affected (*Figure 2H*). This loss of fitness was linked to an intracellular replication defect (*Figure 2I*). In addition to a severe defect in apicoplast inheritance, MyoF deletion resulted in daughter cells growing in abnormal orientations and the formation of enlarged RB filled with secretory organelles (*Jacot et al., 2013*). In FRM2-KO, the orientation of the daughter cells was also altered with increased up/down and down/down orientations in contrast to wt parasites where daughter cells predominantly grew side-by-side toward the apical end (up/up) (*Figure 2—figure supplement 2C*).

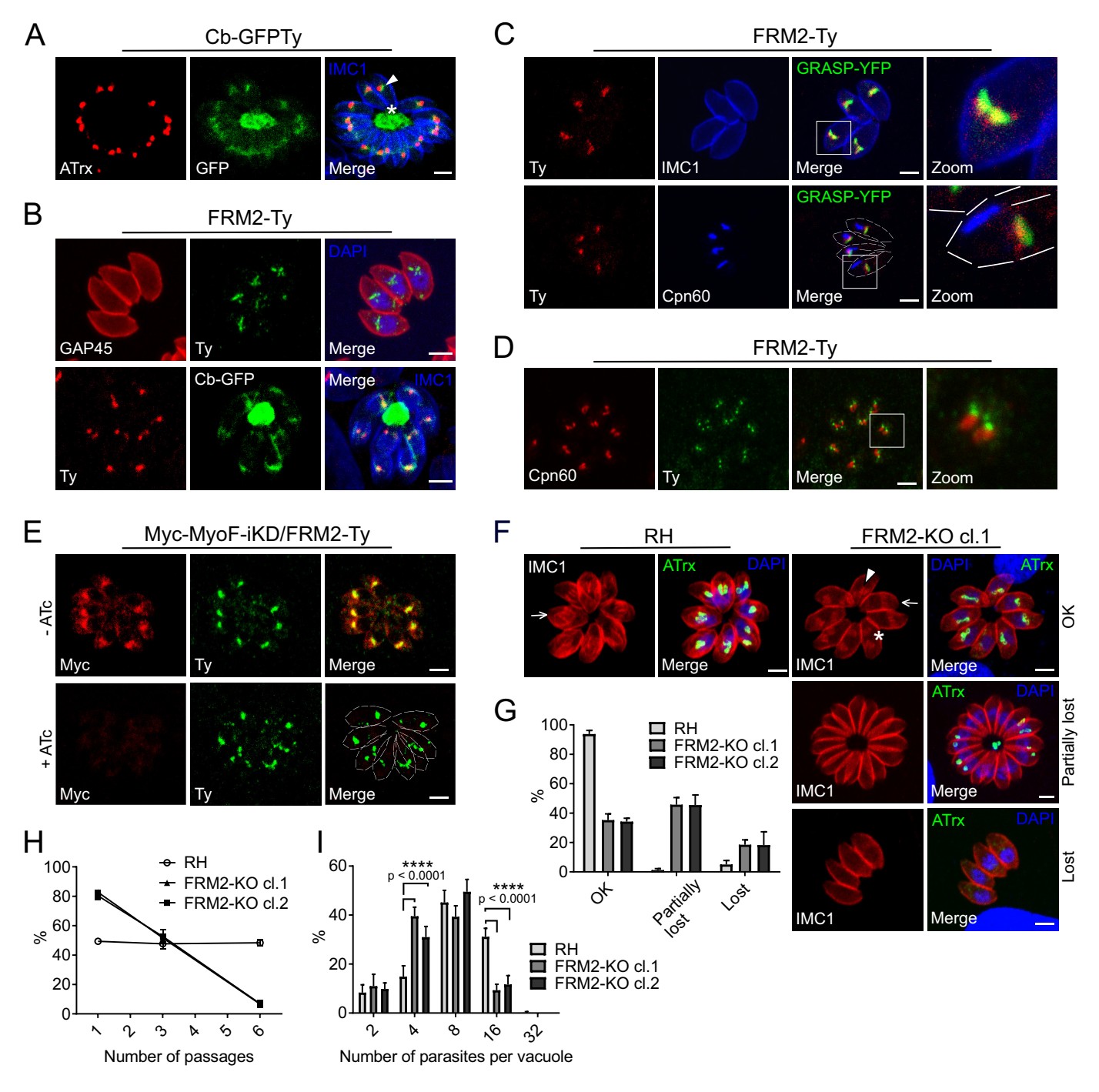

**Figure 2.** FRM2 localizes in a juxtanuclear region and participates in apicoplast inheritance. (A) Expression of Cb-GFPTy in RH parasites showed a strong staining in the RB (asterisk) and in a juxtanuclear region (arrowhead) often overlapping the apicoplast (α-ATrx). α-IMC1 antibodies stain the pellicle. (B) FRM2-Ty is mainly confined above the nucleus and co-localizes with the apical juxtanuclear staining of Cb-GFP. α-GAP45 antibodies stain the pellicle. (C) FRM2-Ty localizes at the proximity of the Golgi (transiently transfected with GRASP-YFP). Triple colocalization of FRM2-Ty, GRASP-YFP and apicoplast (α-Cpn60) showed a constant association of FRM2-Ty with the Golgi, but not with the apicoplast. (D) During daughter cells development, FRM2-Ty accumulates on top of the dividing apicoplast. (E) MyoF (Myc-MyoF-iKD) partially co-localizes with FRM2-Ty and its conditional depletion did not impact on FRM2-Ty localization. (F) Parasites lacking FRM2 are impaired in apicoplast inheritance and showed abnormal daughter cell orientation (arrow: up/up; asterisk: up/down, arrowhead: down/down). (G) Quantification of apicoplast inheritance defects. (H) Growth competition assay reveals a significant defect confirmed by (I) intracellular growth assay. Data are presented as mean ±SD. Significance was assessed using a parametric paired t-test and the two-tailed p-values are written on the graphs. Dashed lines outline parasites periphery. Scale bars: 2 μm.

*Figure 2 continued on next page*

*Figure 2 continued*

DOI: https://doi.org/10.7554/eLife.42669.003

The following source data and figure supplements are available for figure 2:

**Source data 1.** Numerical data of the graphs presented in *Figure 2G, H and I* and *Figure 2—figure supplement 2C*.
DOI: https://doi.org/10.7554/eLife.42669.006
**Figure supplement 1.** FRM2 localization.
DOI: https://doi.org/10.7554/eLife.42669.004
**Figure supplement 2.** Characterization of FRM2-KO.
DOI: https://doi.org/10.7554/eLife.42669.005

The apicoplast and few rhoptries but not micronemes accumulated in the RB (*Figure 2F* and *Figure 2—figure supplement 2D*). Collectively, FRM2 acts in concert with MyoF to ensure apicoplast inheritance.

## FRM3 localizes to the residual body and contributes to cell-cell communication

FRM3 was C-terminally Ty-tagged at the endogenous locus and found weakly expressed, primarily at the basal pole and in the RB (*Figure 3A*). During parasite division, FRM3 was also detected at the tip of nascent daughter cells but in a distinct localization from FRM2 (*Figure 3A* and *Figure 3—figure supplement 1*). Disruption of *FRM3* was performed by double homologous recombination in RHΔKu80 strain and led to no apparent phenotype by plaque assay or competition assay (*Figure 3—figure supplement 2*) as previously described with an independent FRM3-KO mutant (*Daher et al., 2012*). However, the absence of FRM3 also led to an asynchronous division of intravacuolar parasites that was not reported previously and which is indicative of a cell-cell communication defect (*Figure 3B–C*). To directly assess the connection between intravacuolar parasites, fluorescence recovery after photobleaching (FRAP) was carried out following transient transfection of a GFP expressing vector. One or more parasites per vacuole were bleached and the recovery of fluorescence was measured. As shown before (*Frénal et al., 2017b*; *Periz et al., 2017*), fluorescence recovery in RHΔKu80 strain was very fast, reaching a plateau already after ~1 min, which coincided with a concomitant decrease of fluorescence recorded in the neighboring parasites due to the free diffusion of GFP (*Figure 3D*). As control, an entire vacuole was bleached and no fluorescence was recovered even after more than 2 min (*Figure 3—figure supplement 3A*). FRM2-KO showed no defect in fluorescence recovery, ruling out a participation of this actin nucleator in cell-cell communication (*Figure 3—figure supplement 3B–C*). In contrast, parasites lacking FRM3 failed to recover fluorescence or only recovered it at a very slow rate (*Figure 3E–F*). When bleaching two or more parasites, some of them recovered fluorescence more rapidly and the concomitant decrease in fluorescence in the neighboring parasites was heterogeneous. This indicated the existence of few remaining connections between FRM3-KO parasites suggesting that some residual F-actin is sufficient to maintain a limited communication (*Figure 3—figure supplement 3D*).

Synchronized division has recently been linked to MyoI and MyoJ and shown to be dependent on actin (*Frénal et al., 2017b*; *Periz et al., 2017*). MyoI was C-terminally tagged at the endogenous locus in FRM3-KO parasites and shown to be no longer present in the RB but still accumulated at the basal pole (*Figure 3G*). In contrast, the localization of C-terminally tagged MyoJ was not affected and the basal pole constriction appeared normal (*Figure 3H*). The positioning of micronemes and rhoptries, the morphology of the mitochondrion, the inheritance of the apicoplast and the duplication of the Golgi were unaffected by the absence of FRM3 (*Figure 3—figure supplement 4*). Collectively, FRM3 fulfills a function distinct of FRM2 and acts in concert with MyoI to ensure cell-cell communication.

## FRM1 is essential for and exclusively dedicated to gliding motility, invasion and egress

To decisively assess the role of FRM1 (*Daher et al., 2010*), either a single gRNA or a two-gRNA CRISPR/Cas9 approach were designed to disrupt the gene (*Figure 4—figure supplement 1A*). Despite multiple attempts, we failed to remove the entire *FRM1* locus using the two-gRNA approach

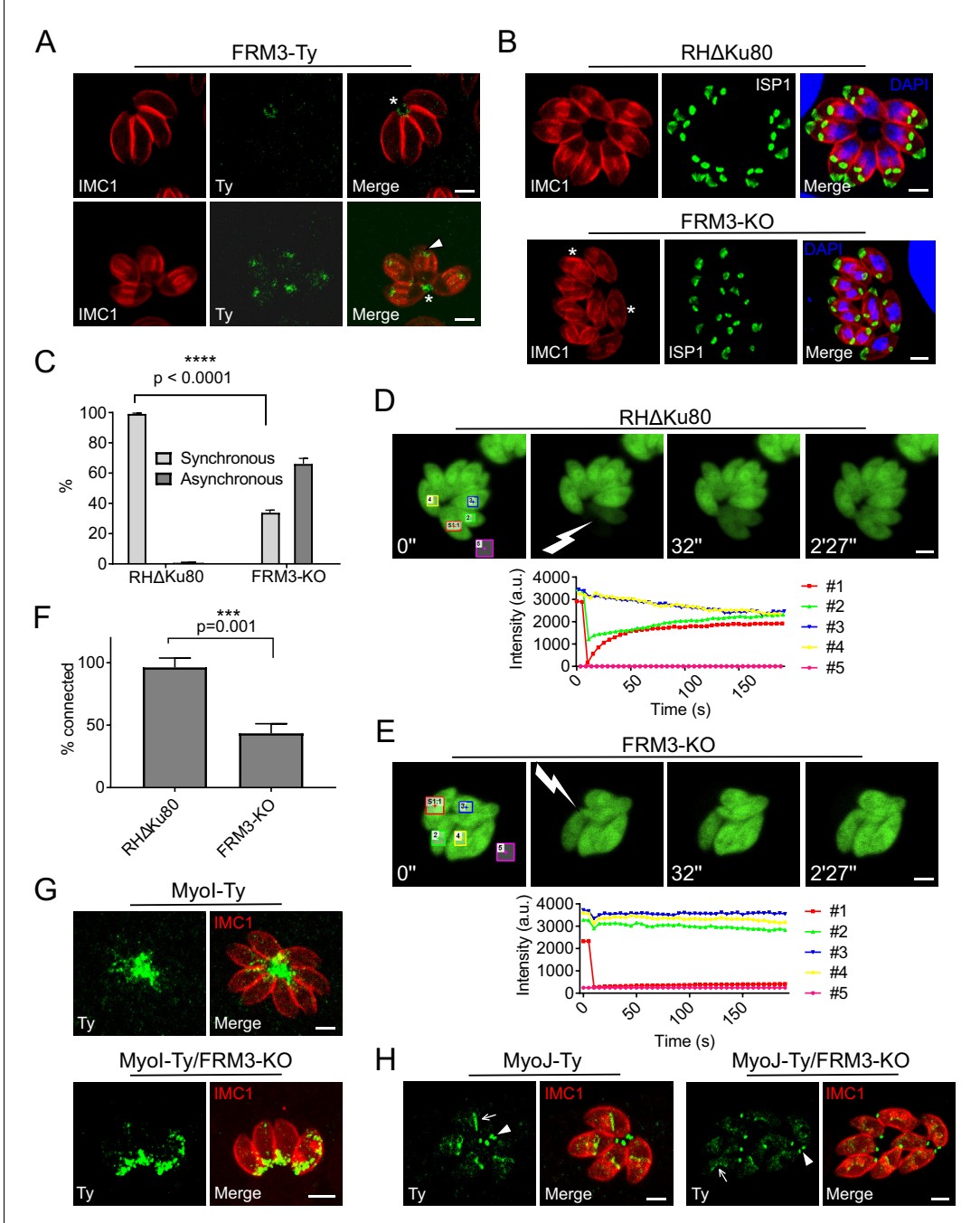

**Figure 3.** FRM3 localizes to the basal pole and residual body and participates in cell-cell communication. (A) FRM3-Ty accumulates at the basal pole and in the residual body (asterisks). FRM3-Ty is also located in the apical region of growing daughter cells (arrowhead). (B) FRM3-KO parasites were unable to form rosettes and divided asynchronously (asterisks). IMC Sub-compartment Proteins 1 (ISP1) stains the apical cap. (C) Quantification of asynchronous division within a vacuole in FRM3-KO parasites. (D–E) Time-lapse images of FRAP experiments in wt RHΔKu80 and FRM3-KO parasites. A flash indicates the bleached area and fluorescence recovery quantifications were recorded in the areas delimited with colors. (F) Quantification of cell-cell communication in wt and FRM3-KO. (G) In FRM3-KO parasites, MyoI is no longer present in the RB. (H) No difference in MyoJ localization or basal pole (arrowheads) constriction was observed in the absence of FRM3. Arrow highlight the basal pole of forming daughter cells. Data are presented as mean ±SD. Significance was assessed using a parametric paired t-test and the two-tailed p-values are written on the graphs. Scale bars: 2 μm.

DOI: https://doi.org/10.7554/eLife.42669.007

The following source data and figure supplements are available for figure 3:

**Source data 1.** Numerical data of the graphs presented in *Figures 3C* and *2F*, and *Figure 3—figure supplement 2C and D*.

DOI: https://doi.org/10.7554/eLife.42669.012

*Figure 3 continued on next page*

*Figure 3 continued*

**Figure supplement 1.** Localization of FRM3.
DOI: https://doi.org/10.7554/eLife.42669.008
**Figure supplement 2.** Generation and characterization of FRM3-KO.
DOI: https://doi.org/10.7554/eLife.42669.009
**Figure supplement 3.** FRAP experiments in FRM2-KO and FRM3-KO.
DOI: https://doi.org/10.7554/eLife.42669.010
**Figure supplement 4.** Characterization of FRM3-KO.
DOI: https://doi.org/10.7554/eLife.42669.011

but managed to isolate two independent clones using a single gRNA. These two clones contained deletions causing out of frame mutations (*Figure 4—figure supplement 1B*) and were severely impacted in all aspect of motility (*Figure 4—figure supplement 1C–E*) without affecting intracellular replication. Remarkably, reverting parasites rapidly emerged after few passages from the initially clonal FRM1-KO population (*Figure 4A*). Several independent sub-clones were isolated and sequencing of the *FRM1* locus revealed spontaneous mutations leading to the correction of the frame shift induced by the CRISPR/Cas9 approach (*Figure 4—figure supplement 1F*). Taken together, this suggests that FRM1 is crucial for parasite motility and the severity of the phenotype measured in various assays might even be underestimated by the confounding fast emergence of revertants. Neither FRM2 nor FRM3 were apparently able to compensate for the loss of FRM1. To tightly and rapidly control FRM1 level, CRISPR/Cas9-mediated gene editing was combined with the plant-like auxin-induced degron (AID) system (*Brown et al., 2017*). The AID sequence followed by a HA-tag was fused to the C-terminus of FRM1 at the endogenous locus (*Figure 4—figure supplement 2A*). FRM1-mAID-HA localized at the tip of the parasite as previously reported (*Jacot et al., 2016*) and was tightly degraded upon addition of indole-3-acetic acid (IAA, auxin) (*Figure 4B*). Depletion of FRM1 resulted in no lysis plaques (*Figure 4C*) and a complete block in host cell invasion, induced egress and gliding motility (*Figure 4D–F*) while intracellular replication, apicoplast inheritance and cell-cell communication were not affected (*Figure 4—figure supplement 2B–D*). During motility, the basal translocation of adhesins can be mimicked and monitored by incubating extracellular parasites with fluorescence beads and assessing their capping to the basal pole (*Whitelaw et al., 2017*). The capping assay leads to either the absence of beads on the parasites (unbound), the distribution of beads over the whole surface of the parasites (bound), the accumulation of beads at the basal end (capped) or an intermediate situation between bound and capped (bound/capped) (*Figure 4—figure supplement 2E*). Strikingly, depletion in FRM1 resulted in a complete absence of capped-events and a large increase of bound-events. The same result was observed upon GAC depletion with however a large increase of incomplete capping (*Figure 4G*). This difference might be explained by the incomplete knockdown of the abundant GAC protein.

FRM1 is firmly anchored to the conoid already in growing daughter cells (*Figure 4H* and *Figure 4—figure supplement 3*), and this tight association is resistant to cytochalasin D (CytD) treatment and to jasplakinolide (JAS), known to cause F-actin apical projections throughout the conoid (*Shaw and Tilney, 1999*) (*Figure 4H*). In conclusion, FRM1 occupies a strategic apical position to solely generate F-actin in order to initiate and sustain motility during egress, gliding and invasion.

## The FRMs fulfill non-overlapping tasks

Cb-GFPTy staining identified two actin polymerization centers in intracellular parasites, located at the basal pole and in the juxtanuclear region (*Figure 2A–B*). To assign these centers to specific nucleators, Cb-GFPTy was transiently expressed in parasites lacking the formins. In FRM2-KO parasites, Cb-GFPTy staining in the juxtanuclear region disappeared almost entirely while the dense staining of the RB remained unchanged. Conversely, in FRM3-KO, Cb-GFPTy exhibited a strong cytosolic staining, predominantly in the juxtanuclear region, whereas the signal at the RB was lost (*Figure 5A* and *Figure 5—figure supplement 1A*). Importantly, absence of FRM1, did not affect Cb-GFPTy staining in intracellular parasites (*Figure 5A*). Of relevance, the transient expression of Cb-GFPTy appeared to affect actin dynamics, since some vacuoles displayed an apicoplast inheritance defect (*Figure 5—figure supplement 1B*).

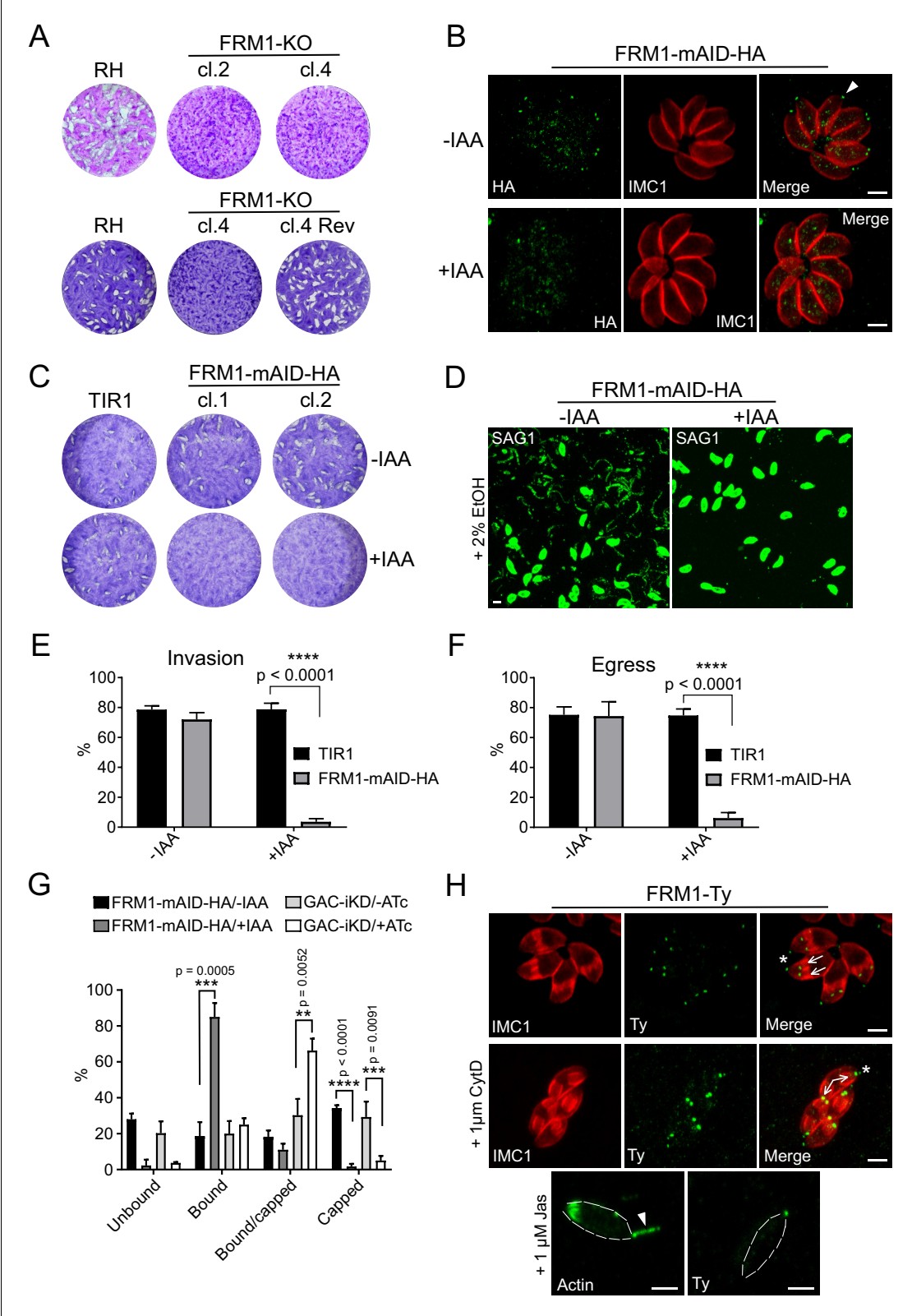

**Figure 4.** FRM1 is localized at the apical tip of parasites to sustain gliding motility, egress and invasion. (A) FRM1-KO resulted in extremely small plaques formed after 7 days compared to RH parasites. Reverted FRM1-KO cl.4 parasites formed plaques comparable to wt parasites. (B) FRM1-mAID-HA localized at the apical tip (arrowheads) and was tightly regulated by IAA. (C) Depletion of FRM1-mAID-HA resulted in no plaques formation after 7 day of IAA treatment. TIR1 represents the parental strain. (D–F) In absence of FRM1 (+IAA), parasites were unable to glide on gelatin-coated glass

*Figure 4 continued on next page*

*Figure 4 continued*

(trails labelled with α-SAG1) and were severely impaired in both egress and invasion. (G) Fluorescent beads capping assay revealed a complete block of capping in absence of FRM1 with a large increase of bound parasites. Conditional depletion of GAC resulted with a block of capping and an accumulation of bound/capped parasites. (H) Localization of FRM1-Ty is restricted to the apical tip of mature parasites (asterisks) and forming daughter cells (arrows). FRM1-Ty localization is not affected upon treatment with CytD and JAS, an actin polymerization enhancer resulting in actin projections (arrowhead). Data are presented as mean ±SD. Significance was assessed using a parametric paired t-test and the two-tailed p-values are written on the graphs. Dashed lines highlight parasites periphery. Scale bars: 2 μm.

DOI: https://doi.org/10.7554/eLife.42669.013

The following source data and figure supplements are available for figure 4:

**Source data 1.** Numerical data of the graphs presented in *Figure 4E, F and G* and *Figure 4—figure supplement 1D and E*.
DOI: https://doi.org/10.7554/eLife.42669.017
**Figure supplement 1.** Generation and characterization of FRM1-KO.
DOI: https://doi.org/10.7554/eLife.42669.014
**Figure supplement 2.** Generation and characterization of FRM1-mAID-HA.
DOI: https://doi.org/10.7554/eLife.42669.015
**Figure supplement 3.** FRM1 associates with the apical end early during division.
DOI: https://doi.org/10.7554/eLife.42669.016

Although FRM2 and FRM3 are restricted to distinct subcellular compartments, the partial phenotypes observed with the individual knockouts compared to the severity of phenotype observed when deleting the corresponding myosins, could be explained by overlapping functions of the two formins. To assess a possible functional redundancy, the double-knockout FRM2/3-KO was generated in RH-ΔKu80-FRM2-Ty parasites by disrupting *FRM2* using a single gRNA and *FRM3* by double homologous recombination (*Figure 5—figure supplement 2A*). Deletion of the two formins resulted in the loss of both the juxtanuclear and the RB F-actin staining (*Figure 5A* and *Figure 5—figure supplement 1A*); however, no further aggravation of the phenotypes by plaque assay, competition assay or in apicoplast inheritance was observed in comparison to the individual knockouts. (*Figure 5—figure supplement 2B–D*). Furthermore, the basal pole constriction still occurred in FRM2/3-KO (*Figure 5B–C*).

Gliding parasites exhibit three distinct forms of motility on glass slides: upright twirling, circular gliding, and helical rotation with only the latter two generating productive movements (*Håkansson et al., 1999*). While FRM1 depletion completely abolished all three types of movement, no motility defect was observed in FRM2/3-KO (*Figure 5—figure supplement 2E*). In conclusion, FRM2 and FRM3 are not implicated in parasite motility, despite the previously suggested involvement of FRM2 obtained from overexpression of the FH2 domain acting as a dominant negative mutant (*Daher et al., 2010*). Actin dynamics was also reported to participate in the trafficking of dense granules (*Heaslip et al., 2016*; *Periz et al., 2017*), possibly favoring their secretion at a still unknown location. Although dense granule movements were not assessed here, FRM2/3-KO displayed normal dense granule proteins accumulation within the PV (*Figure 5—figure supplement 3*) and the nanotubular network appeared unaffected (*Figure 5—figure supplement 4*). Ultimately, to rule-out a possible contribution of FRM1 in FRM2 or FRM3-dependent processes, a knockout of either FRMs was generated in FRM1-AID-HA using a single gRNA CRISPR/Cas9 approach (*Figure 5—figure supplement 5A–B*). Conditional depletion of FRM1 did not aggravate the apicoplast inheritance defect observed in absence of FRM2 (*Figure 5—figure supplement 5C*). Similarly, absence FRM1 in FRM3-KO did not affect basal pole constriction (*Figure 5—figure supplement 5D–E*).

In conclusion, FRM2 and FRM3 play distinct and non-overlapping roles in intracellular parasite, by generating F-actin to sustain MyoF and MyoI function, respectively. In contrast, FRM1 is a nucleator of F-actin exclusively dedicated to motility.

## FRM1 produces an apico-basal flux of F-actin essential for gliding and invasion

To visualize F-actin in moving parasites, Cb-GFPTy expressing wt parasites were stimulated with BIPPO, an inhibitor of cAMP and cGMP phosphodiesterases (PDEs), which activates PKG-dependent egress in *T. gondii* (*Howard et al., 2015*). Remarkably, BIPPO caused a rapid accumulation of F-actin

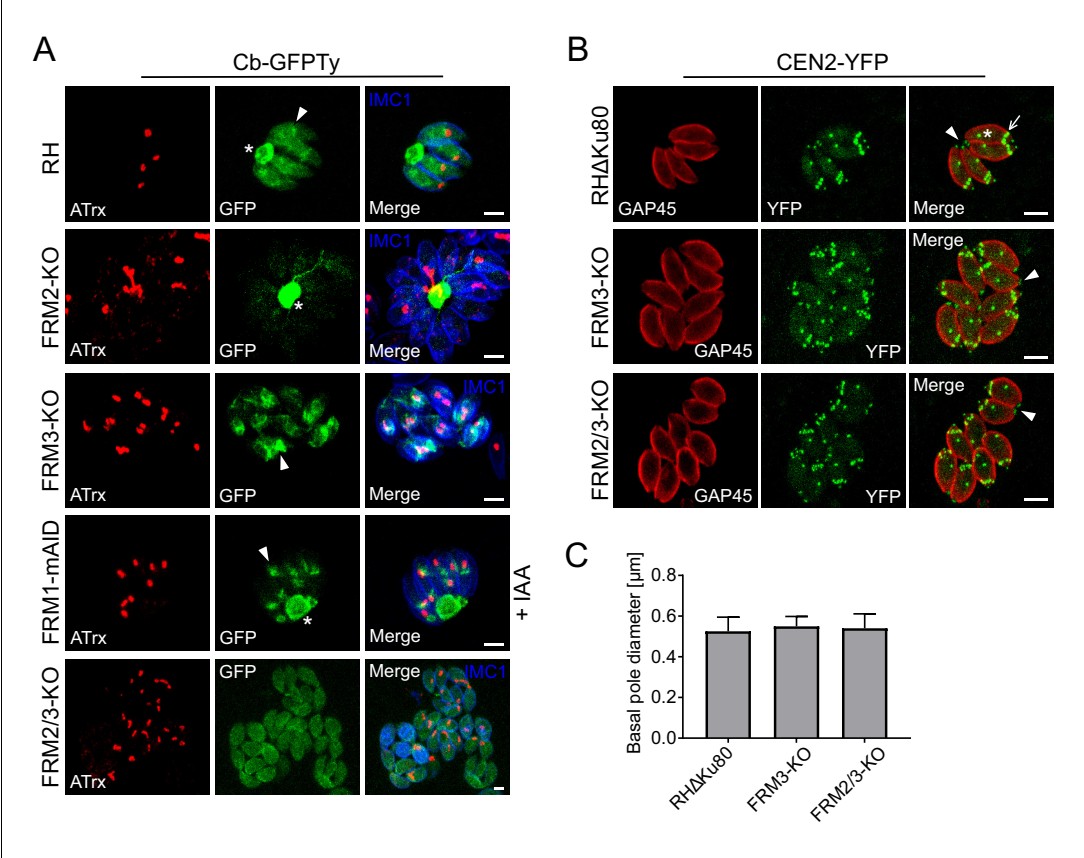

**Figure 5.** The FRMs have no overlapping functions and FRM2 and FRM2 generate the two specific subpopulations of F-actin observed in intracellular parasites. (**A**) Selective disruptions of F-actin staining in the different FRMs knockout. FRM2 is linked to the juxtanuclear Cb-GFPTy staining (arrowhead) while FRM3 generates the F-actin in the RB (asterisks). Conditional depletion of FRM1 was not affecting Cb-GFPTy staining. Absence of both FRM2 and 3 resulted with a diffuse Cb-GFPTy staining. Cb-GFPTy was stably expressed in RH and FRM2/3-KO and transiently transfected in FRM2-KO, FRM1-mAID-HA and FRM3-KO. (**B–C**) Basal pole (arrowheads) constriction is not affected upon deletion of FRM3 or FRM2/3. The EF-hand-containing protein centrin 2 (CEN2) was used as marker of the basal pole and C-terminally YFP tagged at the endogenous locus. CEN2-YFP localizes not only to the basal pole but also to the apical end and annuli (arrows), and to the centrosome (asterisks). Data are presented as mean ±SD. Scale bars: 2 μm.

DOI: https://doi.org/10.7554/eLife.42669.018

The following source data and figure supplements are available for figure 5:

**Source data 1.** Numerical data of the graphs presented in *Figure 5C* and *Figure 5—figure supplements 2C, D and E*, *5C and E*.
DOI: https://doi.org/10.7554/eLife.42669.024
**Figure supplement 1.** Supplementary images of Cb-GFPTy.
DOI: https://doi.org/10.7554/eLife.42669.019
**Figure supplement 2.** Generation and characterization of FRM2/3-KO.
DOI: https://doi.org/10.7554/eLife.42669.020
**Figure supplement 3.** Absence of FRM2 and 3 is not affecting dense granule proteins localizations.
DOI: https://doi.org/10.7554/eLife.42669.021
**Figure supplement 4.** Absence of FRM2 and 3 is not affecting the nanotubular network.
DOI: https://doi.org/10.7554/eLife.42669.022
**Figure supplement 5.** Generations and characterizations of FRM1/2 and FRM1/3 mutants.
DOI: https://doi.org/10.7554/eLife.42669.023

at the basal end of motile wt parasites (*Figure 6A*; *Video 1*). This accumulation appeared very shortly prior to egress and resulted, in less than 30 s, in a Cb-GFPTy staining concentrated as a single dot labeling the basal pole of the parasites. Of relevance, during parasite egress the RB was left behind as a single structure stained with chromobodies. Importantly, the basal accumulation of F-actin was observed in FRM2/3-KO parasites (*Figure 6B*; *Video 2*) ruling-out the participation of

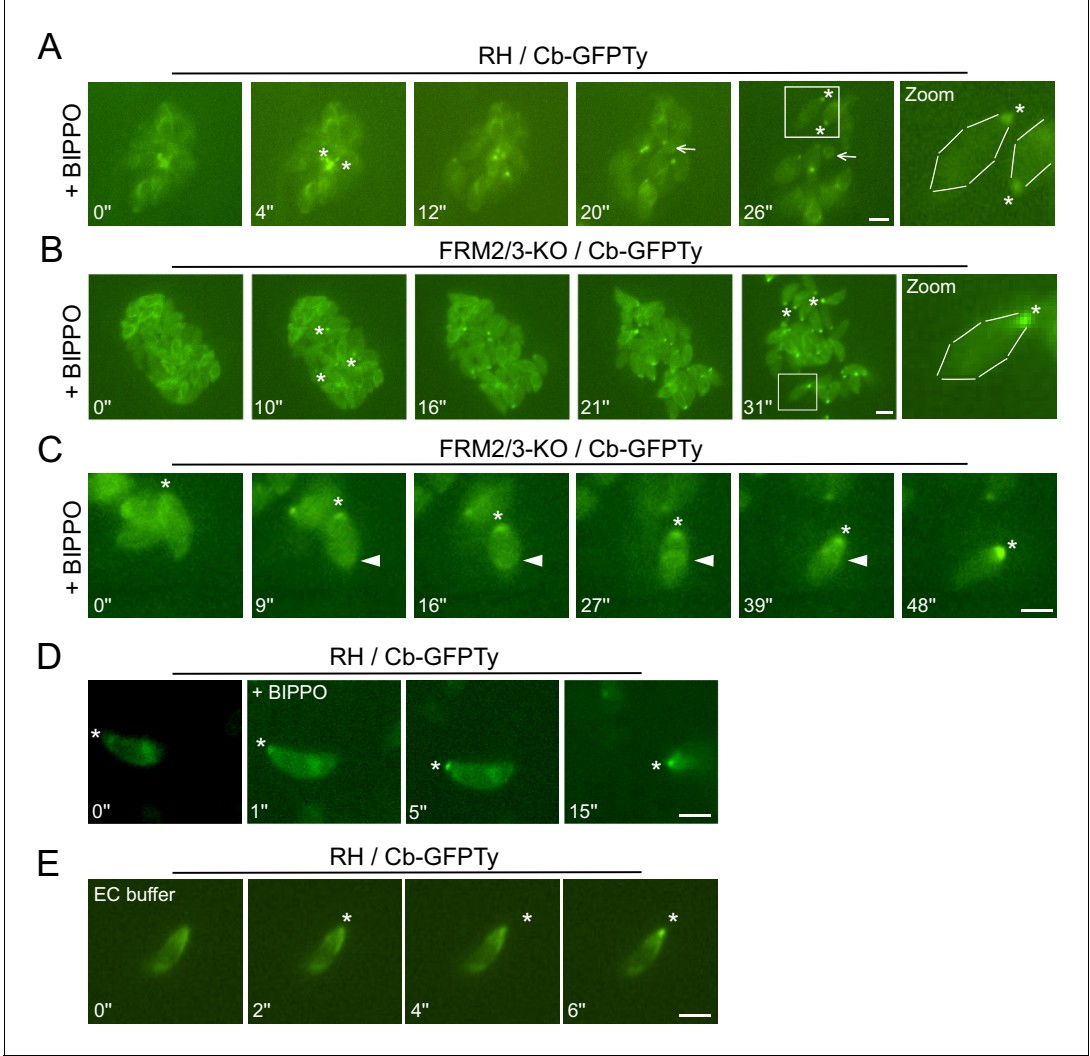

**Figure 6.** Apically generated F-actin by FRM1 accumulates at the basal pole. (A–B) Snapshots of egressing RH and FRM2/3-KO parasites expressing Cb-GFPTy after stimulation with BIPPO. Asterisks represent the accumulation of F-actin at the basal pole. The arrow shows the RB left behind after egress. (C) In invading parasites, a ring of F-actin (arrowheads) translocates from the apical to the basal end of the parasites. (D) Accumulation of F-actin at the basal end (asterisks) was observed prior to parasites movement or (E) even in absence of gliding in extracellular parasites on gelatin coated cover slips. Parasites were either stimulated with BIPPO (responsible for the background change in fluorescence) or incubated in extracellular buffer (EC). Dashed lines highlight parasites periphery. Scale bars: 2 μm.

DOI: https://doi.org/10.7554/eLife.42669.025

FRM2 and FRM3 in this process and leaving FRM1 as the unique actin nucleator implicated in motility.

The restricted presence of FRM1 at the apical tip implies that actin nucleation and polymerization should occur there, followed by the translocation of F-actin to the basal pole (*Graindorge et al., 2016*; *Jacot et al., 2016*; *Long et al., 2017a*). Consistent with the phenotypes collected upon FRM1 depletion, this flux should occur in both invading and gliding parasites. Concordantly, a ring of F-actin translocating from the tip to the basal end was observed in invading parasites resulting in the accumulation of fluorescence at the basal pole (*Figure 6C*; *Video 3*). In extracellular wt parasites stimulated with BIPPO, the basal accumulation of F-actin also occurred but no detectable F-actin staining could be observed either as a ring-like structure or at the point of contact between the parasite and the coated glass. Similarly, F-actin basal accumulation appeared prior to parasite movement (*Figure 6D*; *Video 4*) and even in non-motile parasites (*Figure 6E*; *Video 5*). The same experiments conducted on fixed parasites revealed that Cb-GFPTy co-localized with the MJ in both

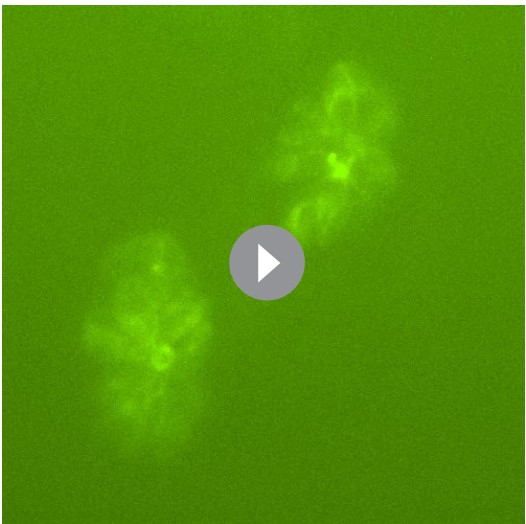

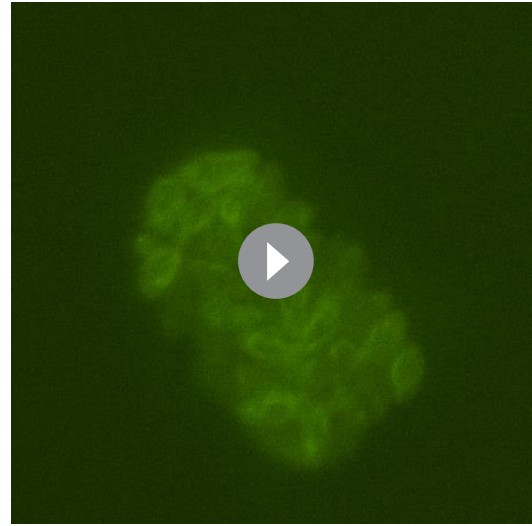

**Video 1.** Progressive basal accumulation of F-actin in RH egressing parasites.
DOI: https://doi.org/10.7554/eLife.42669.026

**Video 2.** Progressive basal accumulation of F-actin in FRM2/3-KO egressing parasites.
DOI: https://doi.org/10.7554/eLife.42669.027

wt and FRM2/3-KO parasites (*Figure 7A*). Accumulation of F-actin at the basal pole occurred in most parasites, labeled and quantified as a single dot of Cb-GFPTy (*Figure 7B–C*). Coronin (COR) is an F-actin-binding protein previously shown to re-localize to the rear of motile parasites both in *T. gondii* and *Plasmodium* (*Bane et al., 2016*; *Salamun et al., 2014*). However, COR is not implicated in the basal accumulation of F-actin (*Figure 7—figure supplement 1*).

To further establish the existence of an apico-basal flow of F-actin in moving parasites, a series of relevant mutants were analyzed. BIPPO stimulated extracellular parasites lacking FRM1 (FRM1-mAID-HA+IAA) showed no accumulation of F-actin in contrast to untreated parasites (*Figure 7D*). Furthermore, BIPPO stimulated extracellular parasites lacking MyoH (MyoH-iKD +ATc) showed no basal F-actin accumulation comforting the view that MyoH translocates F-actin within the pellicular space (*Figure 7E–F*) (*Graindorge et al., 2016*; *Long et al., 2017a*). Remarkably, in MyoA-KO, F-actin accumulated at the junction between the conoid and the IMC (*Figure 7G–H*). Here, MyoH directs F-actin within the pellicle but the flow is blocked at the level of the IMC due to the absence of MyoA. To more directly assess the presence of an apico-basal flux, we performed Reflection Interference Contrast Microscopy (RICM) (*Münter et al., 2009*) on gliding parasites. RICM analyses on circular gliding parasites did not reveal any observable apico-basal signal as the parasites were adhering on their entire length resulting in a continuous signal (*Figure 7I*; *Video 6*). In sharp contrast RICM experiments during helical gliding clearly showed that the parasite first adheres with its apical tip, the adhesion site is then translocated backward while the apical pole detaches from the surface. Concomitant with the adhesion site reaching the basal pole, a second apical adhesion site is generated and starts a second cycle (*Figure 7J*; *Video 7*). Helical gliding is therefore

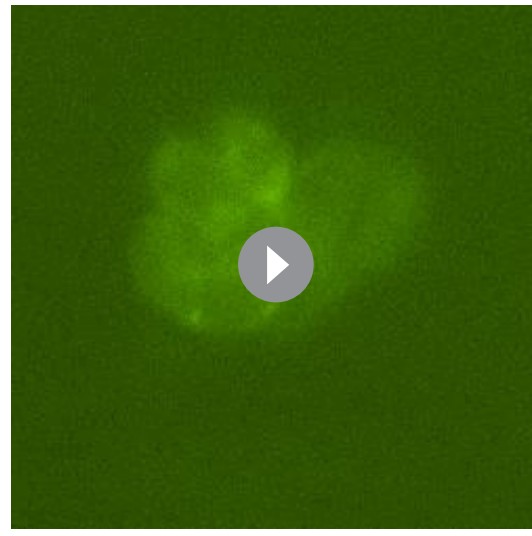

**Video 3.** Ring of F-actin in a moving FRM2/3-KO parasite.
DOI: https://doi.org/10.7554/eLife.42669.031

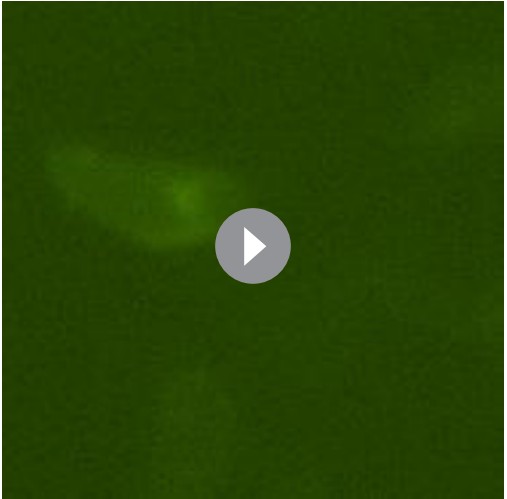

**Video 4.** After BIPPO induction, basal accumulation of F-actin was observed even before parasite movement.
DOI: https://doi.org/10.7554/eLife.42669.032

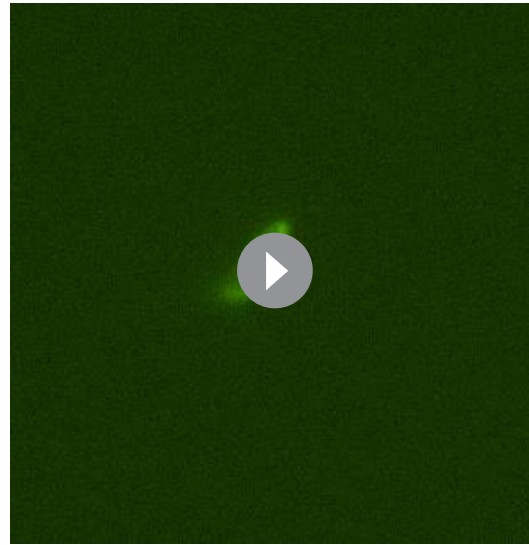

**Video 5.** Basal accumulation of F-actin was observed even without parasite movement. Here extracellular parasites were incubated with extracellular buffer.
DOI: https://doi.org/10.7554/eLife.42669.033

likely composed of successive apico-basal waves of F-actin. Collectively, these data demonstrate that FRM1 generates a flux of F-actin essential to power motility, driven by the successive action of MyoH and MyoA.

## F-actin flux is independent of microneme secretion and controlled by calcium signaling

Due to the essential role of adhesins in motility, it was so far not possible to dissect the signaling pathway leading to actomyosin activation when microneme secretion was concomitantly impaired. To determine whether microneme exocytosis plays a role in the generation of F-actin flux, we took advantage of the transporter TFP1-iKD mutant defective in microneme secretion in presence of ATc (*Hammoudi et al., 2018*). Strikingly, the basal accumulation of Cb-GFPTy was not affected in the absence of TFP1, demonstrating that F-actin flux and microneme exocytosis can be uncoupled and hence offers a unique opportunity to discriminate the signaling events leading to actomyosin system activation and microneme release (*Figure 8A*).

To scrutinize the signaling cascade that governs the F-actin flux, we used selective inhibitors targeting different steps of the pathway. In presence of compound 1 (C1), a potent inhibitor of PKG, BIPPO stimulated extracellular parasites exhibited no basal F-actin accumulation (*Figure 8B*). This formally establishes the central role of cGMP signaling in controlling not only microneme secretion but also actomyosin function. In the current model, PKG is associated to PLC activation and calcium-mediated signaling. PLC produces $IP_3$, which promotes calcium release, while DAG is converted by DGK1 into PA that binds to APH leading to microneme exocytosis (*Bullen et al., 2016*). To discriminate the contributions of the lipid and calcium branches in driving the F-actin flux, extracellular parasites, pre-treated with C1 were stimulated with either calcium ionophore (A23187) or with propranolol, an inhibitor of PA phosphatase (*Bullen et al., 2016*; *Endo et al., 1982*). Only A23187 was able to by-pass the block induced by C1, suggesting an essential contribution of the calcium branch in triggering F-actin flux (*Figure 8B*). This was further confirmed by using the calcium chelator BAPTA-AM, which in addition to blocking microneme secretion completely abolished the flux (*Figure 8—figure supplement 1*). In sharp contrast, the lipid-mediated branch appears to be dedicated to microneme exocytosis only.

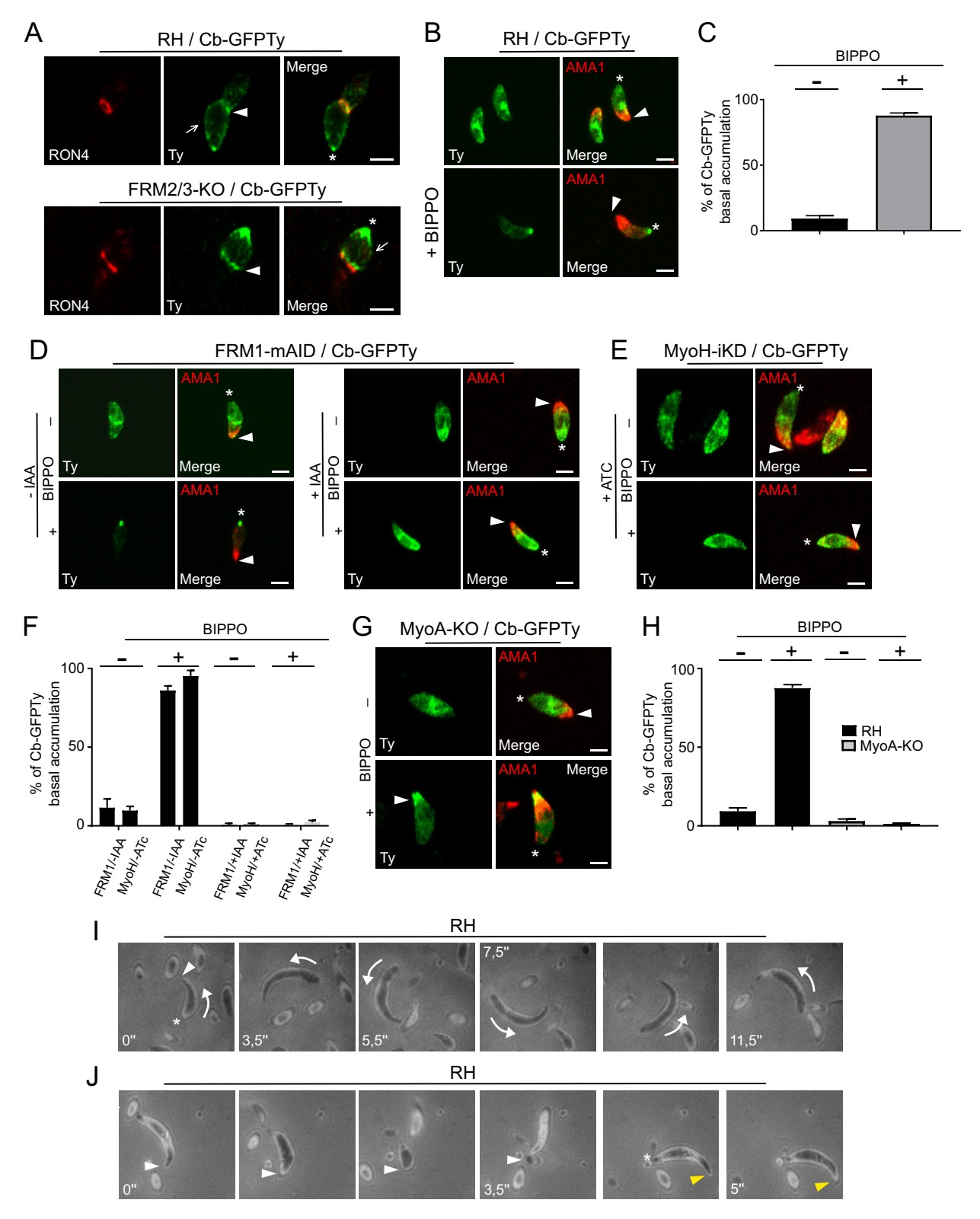

**Figure 7.** An apico-basal F-actin flux is generated by FRM1 and depends on myosins. (**A**) Colocalization of Cb-GFPTy and RON4 at the MJ of invading parasites in wt and FRM2/3-KO parasites (arrowheads). Some F-actin staining can be observed within the pellicle posterior to the MJ (arrows). Asterisks represent the accumulation of F-actin at the basal pole. (**B**) Extracellular wt parasites stimulated with BIPPO, showed a robust accumulation of F-actin by immunofluorescence assays at the basal end with a single basal dot of Cb-GFPTy. α-AMA1 antibodies (arrowheads) label the apical end while asterisks

*Figure 7 continued on next page*

*Figure 7 continued*

show the basal ends. (C) Quantification of basal accumulation of F-actin in (B). (D–F) Contributions of FRM1 and MyoH to the basal accumulation of F-actin in extracellular parasites stimulated with BIPPO. In the absence of MyoH or FRM1, F-actin basal accumulation is abrogated (asterisk). α-AMA1 antibodies label the apical end. (G–H) In absence of MyoA, F-actin accumulates at the start of the IMC (arrowhead). (I) RICM analysis of BIPPO stimulated extracellular parasites. Circular gliding parasites (arrows) were attached on their entire length on the surface resulting in a continuous signal while (J) helical gliding parasites first attached on the surface with their apical end (arrowhead), followed by translocation of the adhesion site backward with a concomitant detachment of the apical end. A second cycle was generated apically (yellow arrowhead) once the adhesion site reached the basal end (asterisk). Data are presented as mean ±SD. Scale bars: 2 μm.

DOI: https://doi.org/10.7554/eLife.42669.028

The following source data and figure supplement are available for figure 7:

**Source data 1.** Numerical data of the graphs presented in *Figure 7F and H* and *Figure 7—figure supplement 1B*.
DOI: https://doi.org/10.7554/eLife.42669.030

**Figure supplement 1.** Basal accumulation of F-actin is independent of COR.
DOI: https://doi.org/10.7554/eLife.42669.029

## Phosphorylation and lysine methylation of proteins control the activation of the actomyosin system

In *T. gondii,* CDPK1 controls the $Ca^{2+}$-dependent secretion of micronemes; however, the molecular mechanism and substrates involved in this process are still elusive (*Lourido et al., 2010*). CDPK3 was also demonstrated to control microneme secretion but only in intracellular parasites and under A23187 stimulation (*Garrison et al., 2012*; *Lourido et al., 2012*; *McCoy et al., 2012*). A CDPK3-KO parasite line was generated (*Figure 8—figure supplement 2A*) and F-actin basal accumulation was assessed on extracellular parasites incubated in intracellular buffer and under calcium stimulation. Under these conditions, microneme secretion was completely blocked as previously reported, but the F-actin flux was not affected (*Figure 8C* and *Figure 8—figure supplement 2B*). Pre-treatment of extracellular parasites with 3MB-PP1, a specific inhibitor of CDPK1 (*Lourido et al., 2012*), resulted in a complete block of F-actin basal accumulation in BIPPO stimulated parasites (*Figure 8D*). This was further confirmed through the generation and analysis of a Tet-repressive conditional knockdown of CDPK1 (CDPK1-iKD) (*Figure 8E* and *Figure 8—figure supplement 3A–B*). As previously reported (*Lourido et al., 2010*), the absence of CDPK1 resulted in a severe defect in the lytic cycle as shown by plaque assay and linked to an impaired microneme secretion (*Figure 8—figure supplement 3C–D*). Further phenotyping of CDPK1-iKD revealed a previously undescribed block in conoid protrusion (*Figure 8F*) that can be assessed by using α-GAC antibodies (*Figure 8—figure supplement 4*) as marker of the protruding organelle in extracellular parasites.

Lysine methylation by AKMT is known to play a critical role in motility without affecting microneme secretion and conoid protrusion (*Heaslip et al., 2011*). Remarkably, depletion of AKMT abrogated the F-actin flux (*Figure 8G*). Since GAC is an F-actin binding and stabilizing protein known to be recruited at the conoid in an AKMT-dependent manner, the role of AKMT in the generation of the flux could have been attributed to the recruitment of GAC. However, the flux of F-actin was still occurring in parasite depleted in GAC (*Figure 8G*). This points to a transport of F-actin along the pellicle, in absence of connection with the adhesins via GAC. Concordantly, F-actin flux occurs in non-motile parasites and in absence of microneme exocytosis. Taken together, the dissection of the F-actin flux demonstrates the unprecedented role of lysine methylation in regulating the actomyosin system, distinct from its participation in the recruitment of GAC to the conoid (*Jacot et al., 2016*).

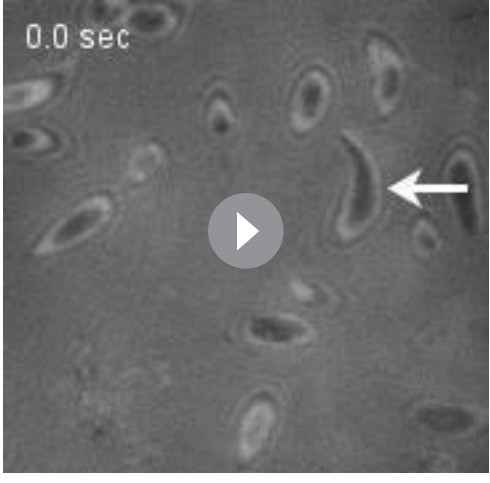

**Video 6.** RICM of circular gliding parasite. Parasites were induced with BIPPO.
DOI: https://doi.org/10.7554/eLife.42669.034

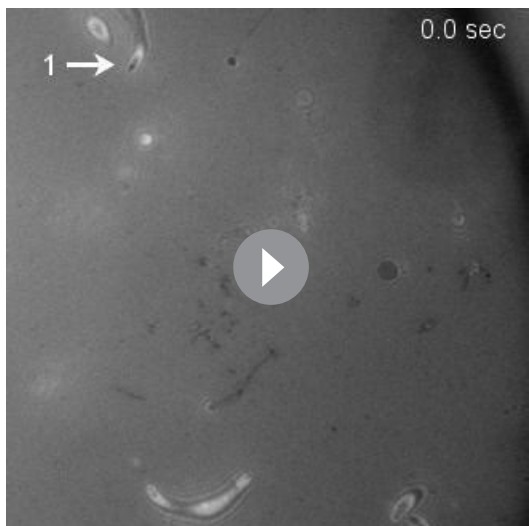

**Video 7.** RICM of helical gliding parasite. Parasites were induced with BIPPO.
DOI: https://doi.org/10.7554/eLife.42669.035

## Discussion

We have established here that the three *T. gondii* formins are responsible for three independent centers of actin nucleation and polymerization (*Figure 9A*). Acting in concert with MyoF, FRM2 is dedicated to the positioning and inheritance of the apicoplast. Distinctly, FRM3 and MyoI work together to enable communication between intravacuolar parasites and their synchronized division. Consistent with this, both FRM3 and MyoI are conserved only in the subgroup of apicomplexans that undergo endodyogeny as well as in *Cryptosporidium* species that divide by merogony (*Mueller et al., 2017*). In contrast, FRM1 is conserved across the phylum, essential for survival, and exclusively dedicated in initiating motility to power invasion and egress. In *P. falciparum*, FRM1 is present at the apical end of merozoites but was additionally localized to the MJ of invading merozoites (*Baum et al., 2008*). FRM1 is also confined at the front of sporozoites and at the poles of gametocytes (*Douglas et al., 2018*; *Hliscs et al., 2015*). On the other hand, FRM2 localizes diffusely throughout the merozoite cytoplasm (*Baum et al., 2008*) a localization compatible with a role in apicoplast inheritance (*Stortz et al., 2018*).

The phenotypes reported upon depletion of either actin-1 (ACT1-cKO) (*Andenmatten et al., 2013*; *Drewry and Sibley, 2015*), the myosin chaperone UNC1 (UNC1-iKD) and each individual myosin (*Frénal et al., 2017b*), are recapitulated by the formins knockouts with only one exception. The basal pole constriction mediated by MyoJ and Centrin 2 (*Frénal et al., 2017b*) still occurred in the absence of individual formins as well as in FRM2/3-KO and FRM1-mAID/FRM3-KO (+IAA); yet this process was shown to be actin-dependent (*Frénal et al., 2017b*; *Periz et al., 2017*). Moreover, the conditional depletion of MyoF, MyoI and MyoJ exhibited more drastic consequences than the double knockouts (*Frénal et al., 2017b*; *Jacot et al., 2013*). Conditional depletion of MyoF resulted in the complete loss of the apicoplast and subsequent parasite death. Deletion of MyoI led to a total block of cell-cell communication, whereas parasites lacking MyoJ were impaired in their basal pole constriction. This led us to conclude that some residual F-actin could still be produced in absence of the formins that would be sufficient to sustain some actin-dependent processes. Polymerization of actin might be provided by a yet unidentified polymerization factor or by the isodesmic properties of actin (*Skillman et al., 2013*). Indeed, polymerization of *T. gondii* ACT1 in vitro was reported to follow an isodesmic model with no lag phase or critical concentration. This could explain the formation of short and heterogeneous filaments in vivo (*Skillman et al., 2013*). In contrast, *P. falciparum* ACT1 appears to polymerize via the classical nucleation-elongation pathway (*Kumpula et al., 2017*).

The extremely severe phenotype observed upon conditional depletion of FRM1 suggests that this sub-population of filaments requires a selective nucleator acting in a temporally and spatially confined and controlled manner. In both gliding and invading parasites, F-actin produced at the apical pole by FRM1 accumulated at the basal pole of the parasite (*Figure 9B*). MyoH ensures the entry of these filaments into the pellicular space and transports them along the protruded conoid to the level of the IMC where MyoA takes the relay for translocation to the basal pole (*Frénal et al., 2014*; *Graindorge et al., 2016*; *Jacot et al., 2016*; *Long et al., 2017a*). In support of this view, the flux of F-actin is dependent on FRM1 and MyoH and in absence of MyoA, F-actin accumulates at the start of the IMC (*Figure 9C*). In light of these observations, FRM1 plays a fundamental role in producing the F-actin flux which upon microneme secretion engages GAC into the glideosome and ensures the translocation of the released pulses of micronemal adhesins toward the basal pole, hence sustaining parasite forward motion. Some data in *Plasmodium* are supportive of this view although the dynamic

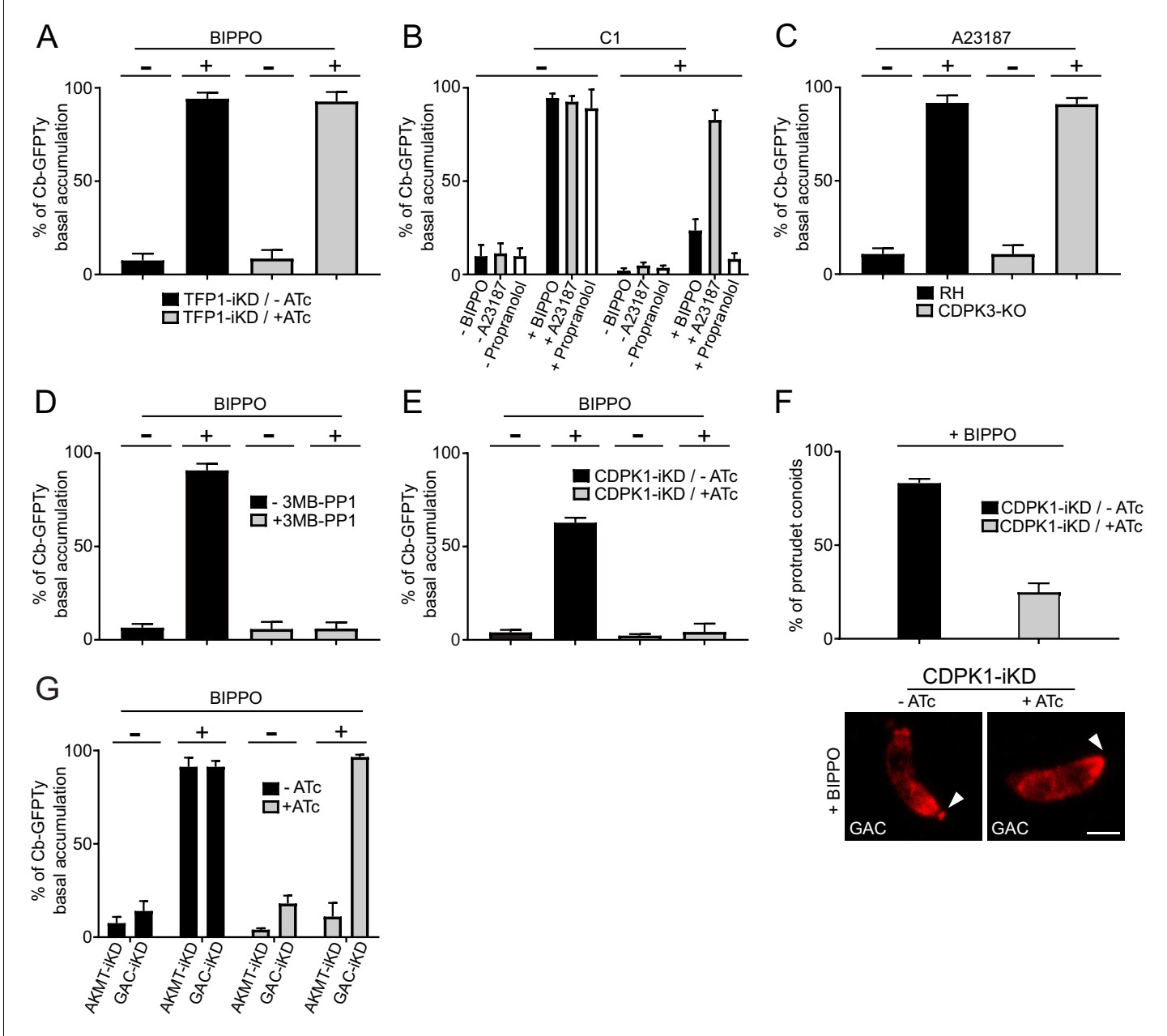

**Figure 8.** Activation of the apico-basal flux of F-actin relies on calcium signaling and AKMT. (A) Absence of microneme secretion, abolished by depletion of TFP1, did not affect the apico-basal flux of F-actin. Extracellular parasites were stimulated with BIPPO. (B) F-actin flux is blocked by C1 (PKG inhibitor) and can only be by-passed with the calcium ionophore A23187. (C) CDPK3 is not involved in F-actin flux. Here, parasites were incubated in intracellular buffer and stimulated with A23187. (D) The CDPK1-specific inhibitor 3MB-PP1 blocked the apico-basal flux of F-actin in extracellular parasites stimulated with BIPPO. (E) Conditional depletion of CDPK1 using the same stimulation, resulted in no F-actin flux and (F) in a severe defect in conoid protrusion (α-GAC arrowheads). (G) AKMT is critical for the establishment of the apico-basal flux, while GAC is dispensable. Scale bar: 2 μm.
DOI: https://doi.org/10.7554/eLife.42669.036

The following source data and figure supplements are available for figure 8:

**Source data 1.** Numerical data of the graphs presented in *Figure 8A, B, C, D, E, F and G* and *Figure 8—figure supplement 1B*.
DOI: https://doi.org/10.7554/eLife.42669.041

**Figure supplement 1.** BAPTA-AM inhibits microneme secretion and F-actin flux.
DOI: https://doi.org/10.7554/eLife.42669.037

**Figure supplement 2.** CDPK3 regulates microneme secretion in intracellular conditions.
DOI: https://doi.org/10.7554/eLife.42669.038

*Figure 8 continued on next page*

*Figure 8 continued*

**Figure supplement 3.** Generation and characterization of CDPK1-iKD.
DOI: https://doi.org/10.7554/eLife.42669.039
**Figure supplement 4.** Validation of GAC antibodies.
DOI: https://doi.org/10.7554/eLife.42669.040

of adhesion sites in gliding sporozoites and ookinete might be different (*Angrisano et al., 2012*; *Kan et al., 2014*; *Münter et al., 2009*; *Quadt et al., 2016*; *Riglar et al., 2011*).

The F-actin flux occurs in non-motile parasites indicating that activation of the actomyosin system occurs independently of GAC engagement with F-actin and the secreted adhesins; two processes required for parasite motility (*Jacot et al., 2016*). In this circumstance, F-actin flows through the pellicle non-productively. This is consistent with the observation that GAC stabilizes pre-existing filaments but is not implicated in de novo nucleation (*Jacot et al., 2016*).

The basal accumulation of F-actin in the absence of microneme exocytosis offered a unique opportunity to identify the signaling cascade controlling the actomyosin system. PKG, which initiates both the calcium and the lipid branches of the signaling pathway, is necessary for both microneme exocytosis and generation of the F-actin flux (*Figure 9D*). However, activation of the actomyosin system appears to depend exclusively on the calcium branch, whereas the lipid branch is dedicated to microneme secretion (*Figure 9D*). The downstream effector of calcium, CDPK3, is not required to generate the F-actin flux, contrasting with previous data suggesting a role of this kinase in the activation of MyoA (*Gaji et al., 2015*; *Tang et al., 2014*). Instead, CDPK1 is instrumental to initiate both the F-actin flux and microneme exocytosis (*Lourido et al., 2010*). Of particular relevance CDPK1 depletion prevents conoid protrusion, an event poorly understood in the context of motility and invasion (*Carey et al., 2004*; *Del Carmen et al., 2009*; *Long et al., 2017b*; *Monteiro et al., 2001*). The conoid protrudes in a calcium-dependent manner and is presumed to be important for microneme secretion. In this context, it is tempting to speculate that conoid protrusion could be a perquisite for the entry of the F-actin flux in the pellicular space. Alternatively, CDPK1 could directly regulate actin polymerization, MyoA or MyoH function.

In *Plasmodium*, activation of PKG is also anticipated to activate PLC and release intracellular calcium ultimately activating several CDPKs (*Brochet and Billker, 2016*; *Fang et al., 2018*; *Gao et al., 2018*; *Moon et al., 2009*). In the blood stage merozoites, CDPK5 appears to control microneme secretion (*Absalon et al., 2018*), while CDPK1 and CDPK4 are implicated in red blood cell invasion possibly through the phosphorylation of proteins of the IMC and glideosome components (*Bansal et al., 2018*; *Fang et al., 2018*; *Green et al., 2008*; *Kumar et al., 2017*). In ookinete, CDPK1 and 4 may play a similar role to support gliding, while CDPK3 seems to be a stage-specific calcium effector controlling gliding through microneme secretion (*Fang et al., 2018*; *Ishino et al., 2006*; *Siden-Kiamos et al., 2006*). Finally, in sporozoites, invasion of hepatocytes requires CDPK4 (*Carey et al., 2014*; *Govindasamy et al., 2016*). Taken together, in both *Plasmodium* and *T. gondii* at least one CDPK is implicated in transducing a calcium signal initiated by PKG into parasite movement. However, the CDPKs appear to have a high degree of functional redundancy making it difficult to assign clearly a single kinase to one function (*Fang et al., 2018*; *Long et al., 2016*). Furthermore, *Plasmodium* does not possess a conoid suggesting that the CDPKs could regulate different processes in these two organisms.

In addition to phosphorylation, lysine methylation is implicated in parasite motility. AKMT was recently linked to the apical recruitment of GAC (*Jacot et al., 2016*) explaining at least in part its mysterious role in motility, invasion and egress (*Heaslip et al., 2011*). However, the critical contribution of AKMT in the generation of the F-actin flux is independent of GAC, pointing to a direct role of this post-translational modification in regulating the actomyosin system. This observation converges with the reported role of AKMT to control the magnitude and polarization of the force driving parasite motion (*Stadler et al., 2017*). AKMT could directly modify actin or modulate the activity of FRM1 and the myosins. Interestingly, in intracellular parasites AKMT is confined at the apical tip, a localization concordant with its role in accumulating GAC at this position (*Jacot et al., 2016*). However, just prior egress, AKMT rapidly re-localizes within the cytoplasm (*Heaslip et al., 2011*) in a process reminiscent of the basal accumulation of F-actin described here. In this context, it is plausible that the dual function of AKMT toward GAC and F-actin might be related to the fluctuating

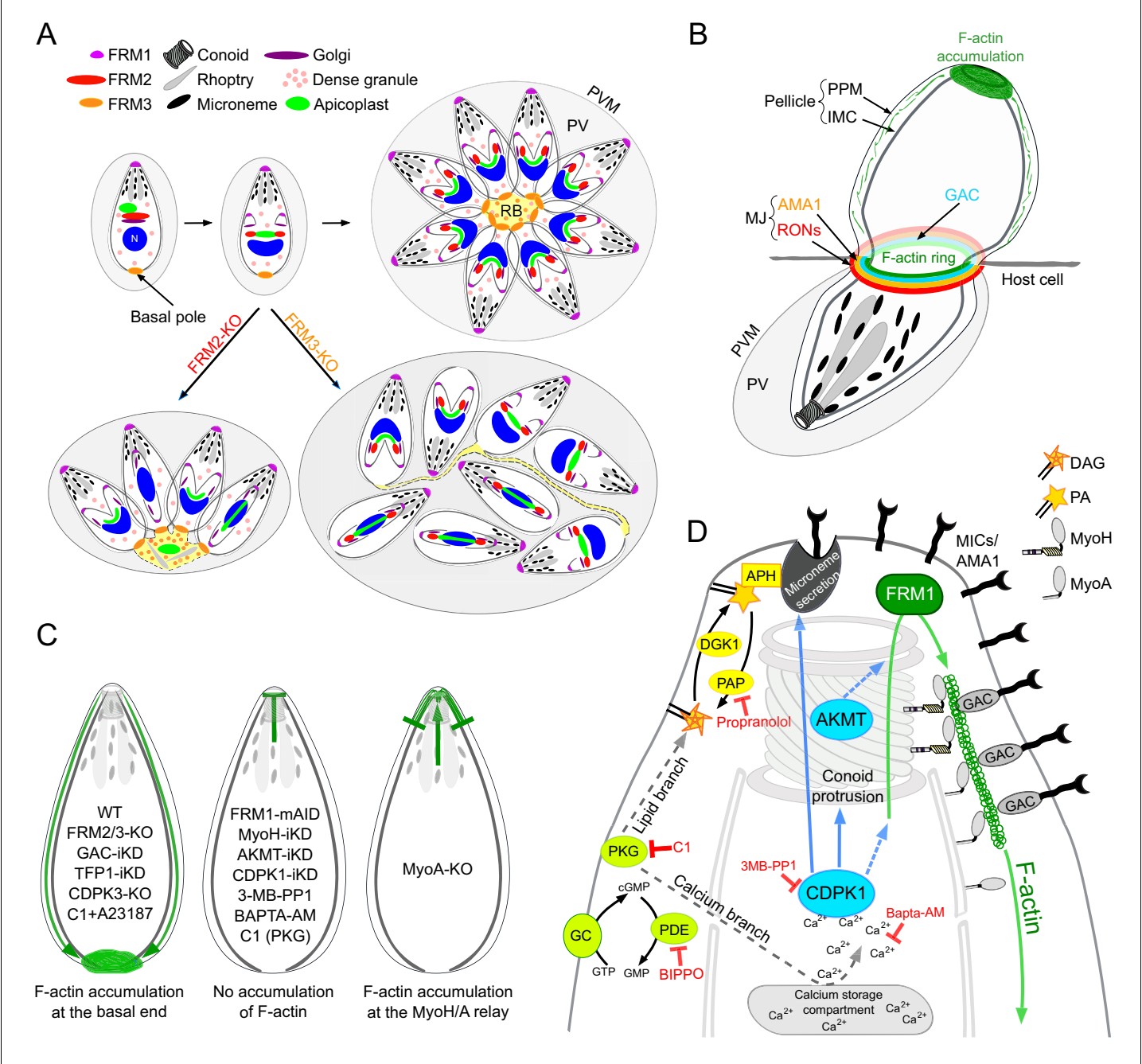

**Figure 9.** Schematic models. (**A**) Schematic representation of the contribution of FRM2 in apicoplast inheritance and FRM3 in synchronous division and rosette formation. (**B**) During invasion, a ring of F-actin translocates with the MJ to the rear of the parasite. Small filaments are likely present within the pellicle and translocated by the actomyosin system to the basal pole, contributing to the F-actin accumulation. (**C**) Schematic summary of the distribution of F-actin produced at the apical end by FRM1 under the different conditions tested in this study. F-actin either accumulated at the basal end (left), did not show any accumulation (middle), accumulated at the junction between MyoH and MyoA (right). (**D**) Schematic summary highlighting the essential roles of myosins, AKMT and calcium signaling in controlling the apico-basal flux of F-actin. The events leading to parasite egress and motility are initiated by the activity of the cGMP-dependent protein kinase (PKG) which activates both the calcium and the lipid branches of the signaling pathway. APH, located on the microneme surface, binds to PA and mediates microneme exocytosis. CDPK1, activated by the calcium release, controls microneme secretion, F-actin apico-basal flux and conoid protrusion. CDPK1 possibly activates AKMT that is also essential for the flux. The exact molecular effectors of both enzymes are however not known. FRM1, localized at the apical end, generates the actin filament that will be further stabilized by GAC in complex with the secreted adhesins. The entire complex will then be translocated to the rear of the parasite, by the successive actions of MyoH and MyoA, generating forward motion.

DOI: https://doi.org/10.7554/eLife.42669.042

localization of this methyltransferase. In *Plasmodium*, the methyltransferase closest to AKMT was localized as distinct foci, apical to the nucleus in erythrocytic and liver stage parasites and throughout the cytoplasm in salivary gland motile sporozoites (*Chen et al., 2016*). The role of this methyltransferase has not been assigned, but the gene is reported to be refractory to knockout in blood stage malaria parasites (*Jiang et al., 2013*).

In conclusion, a shared signaling cascade initiated by cGMP production and PKG activation coordinates microneme exocytosis and F-actin flux. PKG leads to a rise in intracellular calcium that activates CDPK1, a pivotal kinase that controls and synchronizes these two events and potentially also via conoid protrusion. AKMT acts as a key coordinator by enabling the F-actin flux and recruiting GAC (*Figure 9D*). Identification of the range of CDPK1 and AKMT substrates will be instrumental to understand how these posttranslational modifications tightly govern motility, invasion and egress.

## Materials and methods

### Accession numbers

FRM1 (TGME49_206430), FRM2 (TGME49_206580), FRM3 (TGME49_213370), CDPK1 (TGME49_301440), CDPK3 (TGME49_305860)

### Parasite culture

*T. gondii* tachyzoites strains were grown in human foreskin fibroblasts (HFFs, American Type Culture Collection-CRL 1634, absence of mycoplasma contamination was confirmed) maintained in Dulbecco's Modified Eagle's Medium (DMEM, Gibco) supplemented with 5% fetal calf serum (FCS), 2 mM glutamine and 25 µg/ml gentamicin. The RH and a RH strain mutant with Ku80 gene deleted (RHΔKu80) were used as recipient strains. Ku80 is involved in DNA strand repair and non-homologous DNA end joining. In its absence, random integration is eliminated, allowing the insertion of constructs with homologous sequences into the proper loci (*Huynh and Carruthers, 2009*). Depletion of Tet-inducible strains was performed with 1 µg/ml anhydrotetracycline (ATc) (*Meissner, 2001*). Depletion of FRM1-AID-HA was achieved with 500 µM of IAA (*Long et al., 2017a*).

### Cloning of DNA constructs

Genomic DNA was isolated with the Wizard SV genomic DNA purification system (Promega). All amplifications were performed with Q5 (New England Biolabs) polymerase; the primers used are listed in *Supplementary File 1*. All cloning were performed using *E. coli* XL-1 Gold chemo-competent bacteria. To generate the constructs for epitope tagging at the endogenous locus, genomic DNA fragments of the C-terminus of FRM1 (TGME49_206430) and FRM2 (TGME49_206580) were amplified by PCR using primers listed in *Supplementary File 1* (6046/6047; 6090/6091). Vectors were digested with restriction enzymes *ApaI/SbfI* and *ApaI/NsiI*, respectively, and cloned into ASP5-3Ty-DHFR (*Hammoudi et al., 2015*) digested with *KpnI* or *MfeI* and *NsiI*. The FRM3 (TGME49_213370) vector was previously generated (*Daher et al., 2012*). Prior to transfection, the plasmids were linearized with *NsiI*, *MfeI* and AfeI, respectively. Specific gRNA vectors were generated using the Q5 site-directed mutagenesis kit (New England Biolabs) with pSAG1::Cas9-U6::sgUPRT as template (*Shen et al., 2014*). For FRM2-KO, a two gRNAs plasmid was created. Two independent specific gRNA vectors were generated as previously described using primers 6331/4883 and 6537/4883, respectively. A fragment containing the gRNA from the second vector was amplified with Q5 using the primer pair 6147/6148, digested with *KpnI/XhoI* and cloned into the first gRNA vector opened with the same restriction enzymes. FRM3-KO vector was previously made (*Daher et al., 2012*). MyoI, MyoJ, MyoC and Cen2 endogenous tagging vectors were obtained as previously described (*Frénal et al., 2017b*). For Cb-GFP, we amplified the Actin Chromobody (chromotek) from Cb-Halo (*Periz et al., 2017*) by PCR, digested it with *EcoRI/NsiI* and sub-cloned in pT8-NtTgMLC1-GFPTyHXGPRT (*Frénal et al., 2010*) opened with the same enzymes to obtain pT8-Cb-GFPTy-HXGPRT. To create pT8-Cb-GFPTy-CAT, pT8-Cb-GFPTy-HXGPRT was digested with *HindIII/BamHI* and the CAT cassette from pTUB5CATSag1 was inserted using the same restrictions sites. To generate the inducible vector for CDPK1-iKD, a PCR fragment encoding the TATi trans-activator, the HXGPRT cassette and the TetO7S1 promoter was generated using the KOD DNA polymerase (Novagen, Merck) with the vector 5'MyoF-TATi1-HX-tetO7S1MycNtMyoF (*Jacot et al., 2013*) as

template and the primers 8045/8046 that also carry 30 bp homology for double homologous recombination. To direct the insertion of the PCR product at the start of CDPK1, a specific sgRNA vector was generated as described above using the primer pair 8043/4883. KO of CDPK3 was generated following the same strategy as for CDPK1-iKO but KOD PCR were performed on p2854_DHFR-TS using primers 7826/7827 carrying 30 bp homology for double homologous recombination. A specific sgRNA vector was generated using primers 7825/4883. FRM1-mAID was generated following the protocol described in *Brown and Sibley (2018)* using primers 7586/7587 for the KOD PCR and primers 7585/4883 to generate the gRNA. The plasmid pT8-GRA16-3Myc was previously generated (*Hammoudi et al., 2015*).

## Parasite transfection and selection of stable transgenic parasites

*T. gondii* tachyzoites were transfected by electroporation as previously described (*Soldati and Boothroyd, 1993*). Mycophenolic acid (25 mg/mL) and xanthine (50 mg/mL) or pyrimethamine (1 μg/ml) or chloramphenicol (20 mM) or phleomycin (5 mg/ml) were used to select resistant parasites carrying the HXGPRT, the DHFR, the CAT or the Bleo cassettes, respectively. FRM1-KO and FRM2-KO were generated in RH. 48 hr after transfection of 15 μg of the specific gRNA, parasites were GFP-FACS sorted and cloned in 96-well plates. FRM1 mutations were screened by genomic PCR with the primers 6405/6406. Primers 2083/2997 and 2992/2997 were used to check deletion of *FRM2* gene. FRM3-KO was generated as previously described (*Daher et al., 2012*). A second FRM2-KO was generated using a single gRNA approach to disrupt the *FRM2-Ty* locus. The specific gRNA vector was generated as previously described using primers 6331/4883. Disruption of the *FRM2-Ty* locus was assessed by immunofluorescence assay using anti-Ty antibodies. A third FRM2-KO was generated in FRM1-mAID-HA using the same strategy. Disruption of the *FRM3* locus in FRM2-Ty-KO to generate the FRM2/3-KO was generated as described above. A second FRM3-KO was generated using a single gRNA approach in FRM1-mAID-HA. The specific gRNA vector was generated as previously described using primers 7750/4883. PCR analyses were used to assess integration. pT8-Cb-GFPTy-HXGPRT or pT8-Cb-GFPTy-CAT were stably expressed in RH, FRM2-KO and FRM2/3-KO. In all other strains and experiments, either vectors were transfected transiently in the parasites.

## Antibodies

The antibodies used in this study are the following: rabbit polyclonal: α-GAP45 (*Plattner et al., 2008*), α-IMC1 (*Frénal et al., 2014*), α-Cpn60 (*Agrawal et al., 2009*), α-HSP70 (*Pino et al., 2010*), α-ARO (*Mueller et al., 2013*), α-GAC, α-Centrin1 (Kerafast), α-Ty and α-Myc (gifts from Chris Tonkin, WEHI). Mouse monoclonal: α-ACT (*Herm-Götz et al., 2002*) α-ATrx (*DeRocher et al., 2008*), α-ISP1 (*Beck et al., 2010*) α-Ty (BB2), α-Myc (9E10), α-SAG1, α-MIC2, ROP2-4 (gifts from J-F Dubremetz, Montpellier), acetylated α-tubulin (6-11B-1; Santa Cruz Biotechnology). Rat α-HA (3F10, Roche). For immunofluorescence assays, the secondary antibodies Alexa Fluor 405-, Alexa Fluor 488-, Alexa Fluor 594-conjugated goat α-mouse, α-rabbit, or α-rat antibodies (Life Technologies) were used. For western blot analyses, secondary peroxidase conjugated goat α-rabbit or mouse antibodies (Sigma) were used.

## Immunofluorescence assay

Parasite-infected HFF cells seeded on cover slips in 24-well plates were inoculated for 24–30 hr with parasites and fixed with 4% paraformaldehyde (PFA) or 4% PFA/0.05% glutaraldehyde (PFA/GA) in PBS, neutralized in 0.1M glycine/PBS for 3–5 min and processed as previously described (*Plattner et al., 2008*).

## Confocal microscopy and fluorescence recovery after photobleaching (FRAP)

Confocal images were taken with a Zeiss laser scanning confocal microscope (LSM700 using objective apochromat 63x/1.4 oil). Airyscan confocal microscopy was performed with a ZEISS LSM 880 with Airyscan, objective apochromat 63x/1.4 oil. FRAP experiments were conducted as previously described (*Frénal et al., 2017b*). 40 μg of pT8-GFP plasmid was transfected in all reported strains. Experiments were conducted with a Nikon A1r microscope (Ti Eclipse) under stable conditions (37°C; 5% $CO_2$). Acquisitions and processing were done with the software NIS-elements. FRAP

experiments were carried out as follow: initial acquisition step of two images recorded in 1.96 s followed by one bleaches of 7 s with 100% of laser (wavelength 488) and another acquisition step of 3 min with images recorded every 5 s. All experiments were performed at the Bioimaging core facility of the Faculty of Medicine, University of Geneva. Images were processed with ImageJ using maximum intensity Z projection for stacks. FRM1-AID-HA was treated ±IAA 12 hr prior to the assay.

## Electron microscopy

Infected host cells were washed with 0.1 M phosphate buffer pH 7.4 and were fixed with 2.5% glutaraldehyde in 0.1 M phosphate buffer pH 7.4, post-fixed in osmium tetroxide, dehydrated in ethanol and treated with propylene oxide prior to embedding in Spurr's epoxy resin. Thin sections were stained with uranyl acetate and lead citrate. Images were taken with a Technai 20 electron microscope (FEI Company).

## Invasion assay

Extracellular parasites were centrifuged 1 min 1000 rpm and allowed to invade HFF monolayers on coverslips for 30 min before fixation with PFA/GA for 7 min. Fixed cells were indubated during 30 min with 2% BSA/PBS, incubated with α-SAG1 antibodies diluted in 2% BSA/PBS for 20 min and washed three times with PBS. Cells were fixed with 1% formaldehyde/PBS for 7 min and washed once with PBS. Permeabilization using 0.2% Triton X-100/PBS was performed for 20 min. A second incubation using α-GAP45 antibodies diluted in 2% BSA/0.2% Triton X-100/PBS was performed. Cells were washed three times with 0.2% Triton X-100/PBS and incubation with secondary antibodies was performed as described previously. Two hundred parasites were counted for each condition and the percentage of intracellular parasites is represented. Data are mean values ± standard deviation (SD) from three independent biological experiments. FRM1-AID-HA was treated ±IAA 12 hr prior to the assay.

## Egress assay

Freshly egressed tachyzoites were inoculated and grown for 30 hr. The infected host cells were incubated for 7 min at 37°C with DMEM containing either 3 µM of the $Ca^{2+}$ ionophore (A23187) (from *Streptomyces chartreusensis*, Calbiochem) or DMSO as negative control prior to fixation with PFA/GA. Immunofluorescence assays were performed using α-GAP45 antibodies and the average number of lysed vacuoles was determined by counting 200 vacuoles per strain and per condition. Data are presented as mean values ± SD from three independent experiments. FRM1-AID-HA was treated ±IAA 12 hr prior to the assay.

## Gliding assay

Freshly egressed parasites were washed twice with DMEM, resuspended in DMEM supplemented with 2% ethanol and allowed to glide on 24-well plates with gelatin-coated glass slides. The plate was centrifuged 1 min at 1'200 rpm and incubated for 15 min at 37°C and 5% $CO_2$ before fixation with PFA/GA and stained with anti-SAG1 antibodies. One representative data is presented out of three independent experiments. FRM1-AID-HA was treated ±IAA 12 hr prior to the assay.

## Quantification of types of movement in live gliding parasites

Extracellular RH or FRM2/3-KO parasites were placed on gelatin-coated glass and stimulated with BIPPO (5 µM). The type of movement of 100 parasites was scored in three independent experiments. Images were taken by Nikon digital sight camera at 25 frames per second on a Nikon eclipse Ti inverted microscope using a 63 x oil immersion objective.

## Plaque assay

HFFs were infected with freshly egressed parasites and grown for 7 days before fixation with PFA/GA. The host cells monolayer was then stained for 10 min at RT with Giemsa (Sigma-Aldrich GS500). Parasites were treated ±IAA or±ATc from the beginning of the assay. One representative data is presented out of three independent experiments.

## Competition assay

RH and FRM2-KO parasites were mixed with GFP-expressing parasites. The ratios were determined over six passages by immunofluorescence assays using α-IMC1 antibodies and counting 200 vacuoles. The same procedure was used for RH, FRM3-KO and FRM2/3-KO except that ratios were quantified by FACS. Parasites were labeled with Hoechst prior to FACS counting of 10,000 parasites. Data are presented as mean values ± SD from three independent experiments.

## Intracellular growth assay

FRM2-KO and RH parasites were grown for 30 hr prior to fixation with PFA/GA. Immunofluorescence assays using α-GAP45 antibodies was performed and the number of parasites per vacuole was scored. For each condition, 200 vacuoles were counted. Data are mean values ± SD from three independent biological experiments.

## Daughter cell orientation assay and synchronicity of daughter cells

Immunofluorescence assays were performed with α-ISP1 and α-IMC1 antibodies to assess daughter cells orientation and synchronicity within vacuoles. Data are mean values ± SD from three independent biological experiments.

## Pulse invasion assay

Freshly released parasites were inoculated on HFF, centrifuged for 1 min at 1000 g and allowed to invade for 7 min before fixation with PFA/GA for 10 min. Samples were permeabilized with 0.1% Saponin/PBS for 20 min at RT and stained with α-RON4 in 2% BSA/PBS. Samples were then permeabilized with 0.2% Triton X-100/PBS and immunofluorescence assay was performed using α-Ty antibodies as previously described. Representative data are presented from more than three independent experiments.

## Microneme secretion

CDPK1-iKD pre-treated for 48 ± ATc, CDPK3-KO and RHΔKu80 were harvested by centrifugation and the pellets were washed twice in 37°C pre-warmed intracellular buffer (5 mM NaCl, 142 mM KCl, 1 mM MgCl2, 2 mM EGTA, 5.6 mM glucose and 25 mM HEPES, pH 7.2). For the CDPK1 experiment, the pellets were resuspended in DMEM ± ethanol (2%) or ±A23187 (3 µM). For the CDPK3 experiment, parasites were kept in intracellular buffer and stimulated ±A23187 (3 µM). All parasites were incubated at 37°C for 15 min followed by centrifugation at 1000 g for 5 min at 4°C. Pellets were washed once in PBS, whereas supernatants (SN) were centrifuged once more at 2000 g for 5 min at 4°C to remove residual parasite debris. Pellets and supernatants (SN) were analyzed by Western blot using α-MIC2, α-catalase (CAT) and α-dense granule 1 (GRA1) antibodies. One representative data is presented out of three independent experiments.

## Time-lapse video microscopy

RH or FRM2/3-KO parasites were inoculated in fresh HFFs grown on glass bottom plates. After 24 hr parasites were stimulated with BIPPO (5 µM) (*Videos 1–3*) (*Howard et al., 2015*). For *Video 4*, extracellular parasites on gelatin coated glass were stimulated BIPPO (5 µM). For *Video 5*, parasites on gelatin coated glass were stimulated with extracellular buffer 141.8 mM NaCl, 5.8 mM KCl 1 mM MgCl2 1 mM CaCl$_2$5.6 mM Glucose 25 mM HEPES, pH 7.2 with NaOH). Images were taken by Nikon digital sight camera at 25 frames per second on a Nikon eclipse Ti inverted microscope using a 100 x oil immersion objective. Images were processed using ImageJ. More than 10 independent experiments were performed.

## Apico-basal flux of F-actin

Except for RH, FRM2-KO and FRM2/3-KO where pT8-Cb-GFPTy was stably expressed all other strains were transiently transfected with either pT8-Cb-GFPTy-HXGPRT or pT8-Cb-GFPTy-CAT. Transfections were performed 48 hr before the assay. Extracellular parasites were resuspended in DMEM (except CDPK3 that was resuspended in intracellular buffer), placed on 24-well plates coated with 0.1% gelatin, stimulated with either 5 µM BIPPO or 500 µM of propranolol or 3 µM of A23187, centrifuged for 1 min at 1000 g and incubated for 7 min at 37°C/5% CO$_2$. Parasite were fixed for 10

min with PFA/GA and proceeded for immunofluorescence assays as previously described using α-Ty and α-AMA1 antibodies. 15 min pre-treatment with 0.3 μM C1 or 5 μM 3MB-PP1 was performed on extracellular parasites just prior to the assay. The following strains were treated ±ATc before stimulation: GAC-iKD (48 hr), TFP1-IKD (72 hr), CDPK1-iKD (48 hr) and AKMT (48 hr). FRM1-AID-HA was treated ±IAA 12 hr prior to the assay. Data are presented as mean values ± SD from three independent experiments.

## Conoid protrusion

Extracellular CDPK1-iKD parasites treated ±ATc for 48 hr were placed on 24 well plates coated with 0.1% gelatin, stimulated with either 5 μM BIPPO, centrifuged for 1 min at 1000 g and incubated for 7 min at 37°C / 5% $CO_2$. Parasite were fixed for 10 min with PFA/GA and proceeded for immunofluorescence assays as previously described using α-GAC antibodies. Data are presented as mean values ± SD from three independent experiments.

## RICM

Freshy egressed parasites were placed on four wells glass bottom μ-Slides (Ibidi) coated with 0.1% gelatin and were kept in 37°C / 5% $CO_2$ conditions throughout the acquisition using an atmospheric chamber. Gliding was induced by adding BIPPO (final concentration of 5 μM). The acquisition was performed with an Axio Observer Z1 (Carl Zeiss AG) using a Plan Neofluar 63x/1.25 Oil Ph3 Antiflex objective and a QiClick monochrome CCD camera. Short videos were acquired at 2fps and analyzed using the ImageJ software.

## GAC antibodies

To generate the α-GAC antibodies, the full length protein (*Jacot et al., 2016*) was expressed into *E. coli* BL21 strain, affinity purified on Ni-NTA-agarose beads (Qiagen) according to the manufacturer's protocol under nature conditions and used to immunize two rabbits according to the Eurogentec standard protocol. Specificity of the antibodies were assessed by western blot using GAC-iKD ±ATc.

## Beads translocation assay

Beads translocation assay was adapted from *Whitelaw et al. (2017)*. Briefly, 5 μl of Fluorescent latex beads (FluoSpheres carboxylate-modified microspheres, 0.04 μm, Invitrogen) were diluted in 400 μl of H-H buffer (Hanks Balanced Salt Solution +HEPES 25 mM) and sonicated 2 min (4 times 30 s). The aggregated beads were then eliminated by a short spin (1 min at 6000 g), the supernatant was recovered and left on ice for 30 min. Freshly egressed parasites were harvested, washed once with H-H buffer and finally resuspended in H-H buffer to achieve $10^7$ parasites/ml. Then, 250 μl of parasites were transferred on coverslips coated with 0,1% gelatin and incubated on ice for 20 min. 5 μl of diluted beads were added to 250 μl of H-H buffer (complemented with BIPPO 5 μM) and added to the parasites. Immediately, the coverslips were incubated at 37°C for 15 min. The reaction was stopped by fixing the coverslips with 4% PFA for 10 min. The coverslips were then washed with Glycine/PBS for 10 min and the parasites were permeabilized with PBS-Tx100 0.2% for 10 min. After blocking with PBS-BSA 5%, the parasites were stained with an anti-MIC2 in PBS-BSA 2% without Tx100 and washed three times with PBS. Finally, the parasites were incubated with a secondary antibody (Alexa594 – goat anti-mouse IgG) in PBS-BSA 2%, washed three times with PBS and mounted with DAPI-Fluoromount. GAC-iKD parasites were treated ±ATc for 48 hr, FRM1-AiD-HA were treated ±IAA for 12 hr.

## Acknowledgements

We thank Markus Meissner for providing us a vector coding for F-actin Chromobodies and David Sibley for sharing the AID vectors. We are thankful to David Dubois and Hung Ryan Vuong for their critical reading of the manuscript. This research was supported by the Swiss National Science Foundation 310030B_166678 to DS-F, and by Carigest SA to DS-F. Results incorporated in study received funding from the European Research Council (ERC) under the European Union's Horizon 2020 research and innovation program under Grant agreement no. 695596.

## Additional information

### Competing interests
Dominique Soldati-Favre: Reviewing editor, *eLife*. The other authors declare that no competing interests exist.

### Funding

| Funder | Grant reference number | Author |
|---|---|---|
| Carigest SA | | Dominique Soldati-Favre |
| Swiss National Science Foundation | 310030B_166678 | Dominique Soldati-Favre |
| European Research Council | 695596 | Dominique Soldati-Favre |

The funders had no role in study design, data collection and interpretation, or the decision to submit the work for publication.

### Author contributions
Nicolò Tosetti, Conceptualization, Data curation, Formal analysis, Supervision, Validation, Investigation, Visualization, Methodology, Writing—original draft, Writing—review and editing; Nicolas Dos Santos Pacheco, Conceptualization, Resources, Data curation, Formal analysis, Validation, Investigation, Methodology, Writing—review and editing; Dominique Soldati-Favre, Conceptualization, Data curation, Formal analysis, Supervision, Funding acquisition, Validation, Investigation, Visualization, Methodology, Writing—original draft, Project administration, Writing—review and editing; Damien Jacot, Conceptualization, Resources, Data curation, Formal analysis, Supervision, Validation, Investigation, Visualization, Methodology, Writing—original draft, Project administration, Writing—review and editing

### Author ORCIDs
Dominique Soldati-Favre (iD) https://orcid.org/0000-0003-4156-2109
Damien Jacot (iD) http://orcid.org/0000-0001-9932-5443

### Decision letter and Author response
Decision letter https://doi.org/10.7554/eLife.42669.046
Author response https://doi.org/10.7554/eLife.42669.047

## Additional files

### Supplementary files
• Supplementary file 1. Primers listed in the Materials and methods.
DOI: https://doi.org/10.7554/eLife.42669.043
• Transparent reporting form
DOI: https://doi.org/10.7554/eLife.42669.044

### Data availability
All data generated or analysed during this study are included in the manuscript and supporting files.

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
