## [Decision Letter]

Thank you for submitting your article "Three F-actin assembly centers regulate organelle inheritance, cell-cell communication and motility in *Toxoplasma gondii*" for consideration by *eLife*. Your article has been reviewed by Anna Akhmanova as the Senior Editor, a Reviewing Editor, and three reviewers. The reviewers have opted to remain anonymous.

The reviewers have discussed the reviews with one another and the Reviewing Editor has drafted this decision to help you prepare a revised submission.

Summary:

Actin filaments play an important role in motility, organelle position and inheritance in intracellular parasites. However, the precise functions of different actin polymerization machineries and actin filament populations in these processes have remained incompletely understood. This manuscript provides a nicely illustrated and technically advanced analysis of the roles of three formin (FRM) genes/proteins in *Toxoplasma*. The work corrects and updates previous studies on this topic, and provides a comprehensive and accurate picture of the diverse functions for these actin regulatory proteins. Beyond the generally interesting accomplishment of providing another strikingly clear example of isoform-specific functions for formins, this work provides a better insight into several processes that occur in this clinically-important model system.

Thus, all three reviewers found this an interesting and well-conducted study on the mechanisms of *T. gondii* parasite gliding and actin nucleation in intracellular processes of parasite replication. However, they also raised some issues that should be addressed to further strengthen the study.

Essential revisions:

1) The authors note that basal pole constriction is actin-dependent, but that this process appears normal in FRM2/3-KO parasites. Additionally, they note that depletion of particular myosin isoforms have more severe phenotypes than FRM2/3-KO, and speculate that formin-independent actin assembly may also exist. However, an obvious alternative possibility is that FRM1 contributes to these processes. Therefore, it would also be important to test if actin filaments generated by FRM1 participate in the processes for which FRM2 and FRM3 are thought to provide filaments. This could be addressed e.g. by providing a quantitative measure of how many filaments are formed by the distinct FRMs (by using the mutant strains), at least relative to each other by measuring fluorescent intensity of the chromobody at the distinct locations?

2) Is twirling motility blocked in the FRM2/3 KO, or is it only blocked in the FRM1 KO? This would be easy to test and a valuable addition. Because FRM3 is at the back of the cell and twirling involves pirouetting on the basal end, it might be important for this form of motility.

---

## [Author Response]

Essential revisions:1) The authors note that basal pole constriction is actin-dependent, but that this process appears normal in FRM2/3-KO parasites. Additionally, they note that depletion of particular myosin isoforms have more severe phenotypes than FRM2/3-KO, and speculate that formin-independent actin assembly may also exist. However, an obvious alternative possibility is that FRM1 contributes to these processes. Therefore, it would also be important to test if actin filaments generated by FRM1 participate in the processes for which FRM2 and FRM3 are thought to provide filaments. This could be addressed e.g. by providing a quantitative measure of how many filaments are formed by the distinct FRMs (by using the mutant strains), at least relative to each other by measuring fluorescent intensity of the chromobody at the distinct locations?

We have already shown that FRM1 does not directly contribute to the FRM2- or FRM3-dependent process as depletion in FRM1 does not influence apicoplast inheritance and parasites remain connected. However, we cannot exclude that FRM1 could compensate for the absence of either FRM2 or FRM3. The suggested quantitative approach by measuring fluorescent intensity of the chromobodies at the distinct locations is of interest but we consider this technically extremely challenging and likely inconclusive due to the significant heterogeneity of Cb-GFP signal between parasites.

To rule out a contribution of FRM1 in the FRM2 or FRM3 dependent processes, we have instead generated double mutants of FRM1-mAiD-HA/FRM2-KO and FRM1-mAiD-HA/FRM3-KO. No aggravation of the phenotypes was observed upon conditional depletion of FRM1. The double mutants are presented in Figure 5—figure supplement 5.

2) Is twirling motility blocked in the FRM2/3 KO, or is it only blocked in the FRM1 KO? This would be easy to test and a valuable addition. Because FRM3 is at the back of the cell and twirling involves pirouetting on the basal end, it might be important for this form of motility.

While the absence of FRM1 resulted in a complete block of the three forms of motility, FRM2/3-KO displayed normal circular, helical and twirling movements compared to wild type parasites. In consequence and despite its basal pole localization, FRM3 does not contribute to F-actin generation to support twirling. The data are included in the new Figure 5—figure supplement 2.